# Faster Stochastic Optimization with Arbitrary Delays via Adaptive Asynchronous Mini-Batching

Amit Attia [1]   Ofir Gaash [1]   Tomer Koren [1] [2]

## Abstract

We consider the problem of asynchronous stochastic optimization, where an optimization algorithm makes updates based on stale stochastic gradients of the objective that are subject to an arbitrary (possibly adversarial) sequence of delays. We present a procedure which, for any given $q \in (0,1]$, transforms any standard stochastic first-order method to an asynchronous method with convergence guarantee depending on the $q$-quantile delay of the sequence. This approach leads to convergence rates of the form $O(\tau_q/qT + \sigma/\sqrt{qT})$ for non-convex and $O(\tau_q^2/(qT)^2 + \sigma/\sqrt{qT})$ for convex smooth problems, where $\tau_q$ is the $q$-quantile delay, generalizing and improving on existing results that depend on the average delay. We further show a method that automatically adapts to all quantiles simultaneously, without any prior knowledge of the delays, achieving convergence rates of the form $O(\inf_q \tau_q/qT + \sigma/\sqrt{qT})$ for non-convex and $O(\inf_q \tau_q^2/(qT)^2 + \sigma/\sqrt{qT})$ for convex smooth problems. Our technique is based on asynchronous mini-batching with a careful batch-size selection and filtering of stale gradients.

## 1. Introduction

Stochastic first-order optimization methods play a pivotal role in modern machine learning. Given their sequential nature, large-scale applications employ distributed optimization techniques to leverage multiple cores or machines. Mini-batching (Cotter et al., 2011; Dekel et al., 2012; Duchi et al., 2012), perhaps the most common approach, involves computing several stochastic gradients of a single model across a number of distributed workers, sending them to a server for averaging, followed by an update step to the model. A primary drawback of this method is the synchronization performed by the server, which confines the iteration time to be aligned with that of the slowest machine. Hence, variation in arrival time of stochastic gradients due to different factors such as hardware imbalances and varying communication loads, can dramatically degrade optimization performance.

An approach to mitigate degradation caused by delayed gradient computation is asynchronous stochastic optimization (Nedić et al., 2001; Agarwal & Duchi, 2011; Chaturapruek et al., 2015; Lian et al., 2015; Feyzmahdavian et al., 2016), where stochastic gradients from each worker are sent to a server, which applies them immediately and without synchronization. Thus, an update is executed as soon as a stochastic gradient is received by the server, independent of other pending gradients. A significant challenge in these methods is the use of stale gradients, i.e., gradients that were computed in earlier steps and thus suffer from substantial delays, potentially rendering them outdated. Numerous studies have investigated asynchronous optimization under various delay models, including recent results under constant delay (Arjevani et al., 2020; Stich & Karimireddy, 2020), constant compute time per machine (Tyurin & Richtarik, 2023), and the more general arbitrary delay model (Aviv et al., 2021; Cohen et al., 2021; Mishchenko et al., 2022; Koloskova et al., 2022; Feyzmahdavian & Johansson, 2023), where the sequence of delays is entirely arbitrary, and possibly generated by an adversary. The latter challenging setting is the focus of our work.

Early work on asynchronous stochastic optimization considered constant or bounded delay and showed that the maximal delay only affects a lower-order term in the convergence rates, first for quadratic objectives (Arjevani et al., 2020) and subsequently for general smooth functions (Stich & Karimireddy, 2020). More recent studies in the arbitrary delay model established that a tighter dependence can be obtained, moving from dependency on the maximal delay to bounds depending on the average delay (Cohen et al., 2021; Aviv et al., 2021; Feyzmahdavian & Johansson, 2023) or the number of distributed workers (Mishchenko et al., 2022; Koloskova et al., 2022).

---

[1]Blavatnik School of Computer Science, Tel Aviv University [2]Google Research Tel Aviv. Correspondence to: Amit Attia <amitattia@mail.tau.ac.il>.

*Proceedings of the 42$^{nd}$ International Conference on Machine Learning*, Vancouver, Canada. PMLR 267, 2025. Copyright 2025 by the author(s).

However, existing results for the arbitrary delay model suffer from several shortcomings. First and foremost, while the dependence on the average delay is tight in some scenarios (e.g., with constant, or nearly constant delays), it remains unclear whether better guarantees can be achieved in situations with large variations in delays. Indeed, the average delay is known not to be robust to outliers, which are common in vastly distributed settings, and ideally one would desire to rely on a more robust statistic of the delay sequence, such as the median. Second, no accelerated rates have been established for convex smooth optimization, and many of the existing convergence results, notably in non-convex smooth optimization, require a known bound on the average delay or on the number of workers. Lastly, the design and analysis of asynchronous stochastic optimization methods are rather ad-hoc, necessitating the analysis of methods from scratch for each and every optimization scenario.

## 1.1. Summary of Contributions

In this work we address the shortcomings mentioned above. Our main contributions are summarized as follows:

**Black-box conversion for a given quantile delay.** Our first contribution is a simple black-box procedure for transforming standard stochastic optimization algorithms into asynchronous optimization algorithms with convergence rates depending on the median delay, and more generally—on any quantile of choice of the delay sequence. More specifically, given any $q$ and an upper bound $\tau_q$ over the $q$-quantile delay, and given virtually any standard stochastic first-order optimization method, our procedure produces an asynchronous optimization method whose convergence rate depends on $\tau_q$ rather than on the average delay. When coupled with SGD for non-convex or convex Lipschitz objectives, and with accelerated SGD for convex smooth objectives, we establish state-of-the-art convergence results in the respective asynchronous optimization settings; these are detailed in the third column of Table 1.

The guarantees for non-convex and convex optimization improve upon previous results that depend on the average delay bound (Cohen et al., 2021; Mishchenko et al., 2022; Koloskova et al., 2022; Feyzmahdavian & Johansson, 2023). When considering $q = 0.5$ for example, the improvement follows since the median is always bounded by twice the average delay, but it can also be arbitrarily smaller than the average delay.[1] In Appendix C we further demonstrate cases where the median itself is significantly larger than other, smaller quantiles of the sequence; our procedure is able to provide bounds depending on the latter, provided an

upper bound on the respective quantile delay.

Furthermore, to our knowledge, this result yields the first accelerated convergence guarantee for convex smooth problems with arbitrary delays. The results are established in a fully black-box manner, requiring only the simpler analyses of classical methods instead of specialized analyses in the asynchronous setting. We also note that many more convergence results may be obtained using our procedure with other standard optimization methods, for example using SGD for strongly convex objectives, results with high-probability guarantees and with state-of-the-art adaptive methods such as AdaGrad-norm (Ward et al., 2020; Faw et al., 2022; Attia & Koren, 2023; Liu et al., 2023) and other parameter-free methods (Attia & Koren, 2024; Khaled & Jin, 2024; Kreisler et al., 2024).

**Black-box conversion for all quantiles simultaneously.** Our second main contribution is a more intricate black-box procedure which does not require any upper bound on the delays but instead adaptively and automatically matches the convergence rate corresponding to the best quantile-delay bound in hindsight. The fourth column of Table 1 presents the results of the procedure when coupled with SGD and accelerated SGD.

The results provide the first convergence guarantees for asynchronous non-convex smooth and convex Lipschitz problems that does not require any bound of the average delay (or on the number of machines), and accelerated rates for convex smooth optimization. In addition, we recall that the average and median delays could be overly pessimistic compared to other quantile delays (we discuss this in Appendix C), while our adaptive quantile guarantee automatically adjusts to the best quantile bound in hindsight and therefore avoids suboptimal factors in its convergence rate.

**Experimental Evaluation.** Finally, we demonstrate that coupling SGD with our first proposed black-box conversion improves performance over vanilla asynchronous SGD when training a neural network on a classification task in a simulated asynchronous computation setting.

## 1.2. Discussion and Relation to Previous Work

**Fixed delay model.** Earlier work on delayed stochastic optimization considered a model where the delay is fixed. The work of Arjevani et al. (2020) on quadratic objectives followed by the results of Stich & Karimireddy (2020) for general smooth objectives established that the delay parameter affects the "deterministic" low-order convergence term of SGD, which can be extended to a dependence on the maximal delay in the more general arbitrary delay model. This dependence on the maximal delay is the best one can achieve for (fixed stepsize) vanilla SGD in the arbitrary de-

---

[1] Indeed, this follows from a simple application of Markov's inequality: $\tau_{\mathrm{med}} \le \tau_{\mathrm{avg}}/\Pr(d_t \ge \tau_{\mathrm{med}}) \le 2\tau_{\mathrm{avg}}$, where $t$ is chosen uniformly at random. On the other hand, for the delay sequence $d_t = \mathbf{1}[t > T/2 + 1](t-1)$, we have $\tau_{\mathrm{med}} = 0$ and $\tau_{\mathrm{avg}} = \Omega(T)$.

Table 1: Convergence results for $T$ rounds of asynchronous stochastic optimization with arbitrary delays (other delay models are omitted from the table); $\tau_{\mathrm{avg}}$ is the average delay and $\tau_q$ is the $q$-quantile delay. Note that quantile-delay bounds are superior to average-delay bounds, as the median delay ($q = 1/2$) is always bounded by twice the average delay, and may be significantly smaller; see Appendix C for details. Only the results of Algorithm 2 and (Aviv et al., 2021) depend on the actual delay parameters and not on known upper bounds. The results of Koloskova et al. (2022); Mishchenko et al. (2022) depend in fact on the number of machines in a master-worker setup, which upper bounds the average delay (see Appendix B).

| SETTING | PRIOR STATE-OF-THE-ART | ALGORITHM 1 (COR. 2) | ALGORITHM 2 (THMS. 3,6) | CENTRALIZED OPTIMIZATION |
|---|---|---|---|---|
| non-convex, smooth | $\frac{1+\tau_{\mathrm{avg}}}{T} + \frac{\sigma}{\sqrt{T}}$ [a] | $\frac{1+\tau_q}{qT} + \frac{\sigma}{\sqrt{qT}}$ | $\inf_q \frac{1+\tau_q}{qT} + \frac{\sigma}{\sqrt{qT}}$ | $\frac{1}{T} + \frac{\sigma}{\sqrt{T}}$ [d] |
| convex, smooth | $\frac{1+\tau_{\mathrm{avg}}}{T} + \frac{\sigma}{\sqrt{T}}$ [b] | $\frac{1+\tau_q^2}{(qT)^2} + \frac{\sigma}{\sqrt{qT}}$ | $\inf_q \frac{1+\tau_q^2}{(qT)^2} + \frac{\sigma}{\sqrt{qT}}$ | $\frac{1}{T^2} + \frac{\sigma}{\sqrt{T}}$ [e] |
| convex, non-smooth | $\frac{\sqrt{1+\tau_{\mathrm{avg}}(1+\sigma)}}{\sqrt{T}}$ [c] | $\frac{\sqrt{1+\tau_q+\sigma}}{\sqrt{qT}}$ | $\inf_q \frac{\sqrt{1+\tau_q+\sigma}}{\sqrt{T}}$ | $\frac{1+\sigma}{\sqrt{T}}$ [e] |

[a](Cohen et al., 2021; Mishchenko et al., 2022; Koloskova et al., 2022), [b](Aviv et al., 2021; Cohen et al., 2021; Feyzmahdavian & Johansson, 2023), [c](Mishchenko et al., 2022), [d](Ghadimi & Lan, 2013), [e](Lan, 2012).

layed model, without further modifications to the algorithm; for completeness, we prove this formally in Appendix D.

**Bounded average delay.** Several recent works targeted the arbitrary delay model, devising methods which obtain better convergence rates that does not depend on the maximal delay. Cohen et al. (2021) proposed the "picky SGD" method, where stale gradients are ignored if they are far from the current step, obtaining rates that depends on the average delay for non-convex and convex-smooth problems. Although their method does not implicitly require a known delay bound, to ensure convergence given a fixed steps budget an average delay bound is required. In addition, their method provides a guarantee only for the best encountered point (which cannot be computed efficiently) instead of the output of the algorithm, and non-accelerated rates for convex-smooth objectives. Feyzmahdavian & Johansson (2023) provides a guarantee for convex smooth optimization that depends on the average delay by filtering gradients with delay larger than twice the average delay bound (which is essentially used as a median delay bound). Similarly to the other works, a known bound is required and only a non-accelerated rate is provided. Aviv et al. (2021) is the only work we are aware of that does not require a known delay bound, but their result holds only for convex-smooth problems and does not yield accelerated rates. Our quantile-adaptive results, on the other hand, avoid the limitations mentioned above.

Two later studies focused on the arbitrary delay model but under the assumption of a bounded number of machines (Koloskova et al., 2022; Mishchenko et al., 2022). By adjusting the stepsizes of SGD according to this quantity they provided guarantees that depends on the number of machines. As previously observed, the average delay is in fact bounded by the number of machines (see Appendix B for a short proof) and as such we improve upon these results in a similar fashion.

**Other delay models.** While the arbitrary delay model has garnered attention in recent years, additional delay models were considered in previous work. Sra et al. (2016) studied the convergence of asynchronous SGD while either assuming a uniform delay distribution with a bounded support or delay distributions with bounded first and second moments. Closely related to our paper is the work by Tyurin & Richtarik (2023), who use mini-batching to achieve guarantees for non-convex, convex and convex smooth optimization (included accelerated rates) using standard optimization methods. The asynchronous model of (Tyurin & Richtarik, 2023) is limited to a constant compute time per machine with a finite number of machines, and does not support variation in delays due to variable communication times and failures of machines, like the arbitrary delays does. Our approach may be viewed as a considerable generalization of the asynchronous mini-batching technique to the arbitrary delay model, achieving adaptivity to the delay quantiles instead of the top-$m$ machines as in (Tyurin & Richtarik, 2023). Yet another delay model studied in several works involves delayed updates to specific coordinates in a shared parameter vector (Recht et al., 2011; Mania et al., 2017; Leblond et al., 2018).

**Asynchronous mini-batching.** Several works used mini-batching in an asynchronous setting. Feyzmahdavian et al. (2016) considered the use of mini-batching at the worker level, using a maximal delay bound and achieving suboptimal results with respect to the maximal delay. We on the other hand use mini-batching across workers to reduce the effect of stale gradients. Dutta et al. (2018) discussed several variants of synchronous and asynchronous SGD (including *K-batch sync SGD* which is similar to our use of asynchronous mini-batching), focusing on the wall clock-time given certain distribution assumptions on the compute time per worker. While providing insight regarding the runtime of the variants, they do not consider the fact that one

can increase the batch size of SGD to a certain point (at the cost of fewer update steps) without degrading the theoretical performance. As mentioned above, the closely related "Rennala SGD" method of Tyurin & Richtarik (2023) used asynchronous mini-batching in a more restricted fixed computation model, where each machine $m$ computes gradients at a constant $\Delta_m$ time.

**Beyond homogeneous i.i.d. data.** One way to move beyond the standard assumption of homogeneous i.i.d. data is to consider asynchronous optimization with heterogeneous data, where different workers access disjoint subsets of the training set (Mishchenko et al., 2022; Koloskova et al., 2022; Tyurin & Richtarik, 2023; Islamov et al., 2024). Prior work in this setting often relies on additional structural assumptions–for example, fixed per-machine delays (Tyurin & Richtarik, 2023, Theorem A.4) or specific distributional assumptions (Mishchenko et al., 2022, Assumption 2)–which are typically required to ensure that data on slower machines is accessed frequently enough for effective optimization. Moreover, recent work has explored the impact of Markovian sampling strategies, where temporally correlated samples induce biased gradient oracles, as commonly encountered in reinforcement learning, on optimization with delayed feedback (Adibi et al., 2024; Dal Fabbro et al., 2024). Extending our framework to accommodate such forms of data heterogeneity and temporal dependence presents a promising direction for future research.

**Delayed feedback in online learning and Multi-Armed Bandits.** Scenarios of delayed feedback were also studied in the literature on online learning and multi-armed bandits (Dudik et al., 2011; Joulani et al., 2013; Vernade et al., 2017; Gael et al., 2020). In particular, the work of Lancewicki et al. (2021) achieves a regret guarantee with an adaptivity to the quantiles of the delays distribution, analogous to our result in the stochastic optimization setup.

**Concurrent work.** Recently and concurrently to our work, Tyurin (2025) and Maranjyan et al. (2024) studied delay models that extend the fixed-compute framework of Tyurin & Richtarik (2023). Most relevant to our work, Tyurin (2025) demonstrated that SGD with gradient filtering and accumulation is optimal in their time-delay model, by proving a lower bound for zero-respecting first-order algorithms. While their computational model accommodates arbitrary delays, it deviates from prior asynchronous frameworks (e.g., Cohen et al., 2021; Mishchenko et al., 2022; Koloskova et al., 2022) and results in recursive convergence bounds that lack direct interpretation. In contrast, our approach provides more informative guarantees derived from natural empirical quantities, such as the quantiles of observed delays. Maranjyan et al. (2024), on the other hand, addresses scenarios where worker computations may take

arbitrarily long or hang indefinitely. Notably, their method requires prior knowledge of delay distributions and allowing worker restarts mid-computation.

## 2. Preliminaries

### 2.1. Stochastic Optimization Setup

In this work we are interested in minimization of a differentiable function $f : \mathcal{W} \to \mathbb{R}$ for some convex set $\mathcal{W} \subseteq \mathbb{R}^d$. We assume a standard stochastic first-order model, in which we are provided with an unbiased gradient oracle $g : \mathcal{W} \to \mathbb{R}^d$ with bounded variance, i.e., for any $w$,

$$\mathbb{E}[g(w)] = \nabla f(w) \quad \text{and} \quad \mathbb{E}[\|g(w) - \nabla f(w)\|^2] \leq \sigma^2$$

for some $\sigma \geq 0$. In the following sections we will discuss algorithms which receive an initialization $w_1 \in \mathcal{W}$, performs $T$ gradient queries and output a point $\widehat{w}$. We consider the following optimization scenarios:

(i) ***Non-convex setting.*** In this case we assume that the domain is unconstrained ($\mathcal{W} = \mathbb{R}^d$), $f$ is $\beta$-smooth,[2] and lower bounded by some $f^\star$. We also assume that the algorithm receives a bound $F \geq f(w_1) - f^\star$.

(ii) ***Convex non-smooth setting.*** Here we assume that the domain has a bounded diameter, i.e., for any $x, y \in \mathcal{W}$, $\|x - y\| \leq D$ for some $D > 0$, and that $f$ is convex and $G$-Lipschitz over $\mathcal{W}$.

(iii) ***Convex smooth setting.*** Finally, we also consider the setting where domain is unconstrained ($\mathcal{W} = \mathbb{R}^d$) and the objective $f$ is convex, $\beta$-smooth, and admits a minimizer $w^\star \in \arg\min f(w)$. We also assume that the algorithm receives a bound $D \geq \|w_1 - w^\star\|$.

### 2.2. Asynchronous Optimization with Arbitrary Delays

We consider the same asynchronous optimization model as in (Aviv et al., 2021; Cohen et al., 2021), which consists of $T$ rounds, where each round involves the following: *(i)* The algorithm chooses a model $w_t$; *(ii)* the algorithm receives a pair $(g_t, d_t)$, where $g_t$ is a stochastic gradient of $f$ at $w_{t-d_t}$. The delays $d_t$ here are entirely arbitrary, and can be thought of as chosen by an oblivious adversary (in advanced, before the actual optimization process begins). Note this means that the delays are assumed to be independent (in a probabilistic sense) of the stochastic gradients observed during optimization.

Throughout, we let $\tau_{\text{avg}}$ denote the average delay; $\tau_{\text{med}}$ denote the median delay; and $\tau_q$ denote the $q$-quantile delay[3].

---

[2] A function $f$ is said to be $\beta$-smooth if for any $x, y \in \mathcal{W}$, $\|\nabla f(x) - \nabla f(y)\| \leq \beta\|x - y\|$. This condition also implies that for any $x, y \in \mathcal{W}$, $f(y) \leq f(x) + \nabla f(x) \cdot (y - x) + \frac{1}{2}\beta\|y - x\|^2$.

[3] We define a $q$-quantile of the delay sequence for $q \in (0, 1]$ as

We further treat the quantile delays ($\tau_q$ for $q \in (0, 1]$) as integers. This can be done without loss of generality because the delays are all integers, so $\lfloor \tau_q \rfloor$ remains a valid $q$-quantile, and smaller quantile delays tighten our guarantees.

## 2.3. Classical Stochastic First-Order Optimization

Our asynchronous mini-batching technique enables the direct application of classical stochastic optimization methods in a delayed setting. Since we use these methods in a black-box manner, we do not present them here but rather only state the relevant rates of convergence. The exact convergence results we give here are taken from Lan (2020) and are specified in more detail in Appendix A.

Following are the algorithms we use in our work, which receive an initialization $w_1 \in \mathcal{W}$, performs $T$ gradient queries and output a point $\widehat{w}$:

(i) *Stochastic gradient descent* (SGD, Lan, 2020), which in the non-convex setting guarantees that $\mathbb{E}[\|\nabla f(\widehat{w})\|^2] = O(\beta F/T + \sigma\sqrt{\beta F}/\sqrt{T})$, and in the convex smooth setting provides the guarantee $\mathbb{E}[f(\widehat{w}) - f(w^\star)] = O(\beta D^2/T + \sigma D/\sqrt{T})$ (Lan, 2020, Corollary 6.1).

(ii) *Projected stochastic gradient descent* (PSGD, Lan, 2020), which in the convex non-smooth setting provides a guarantee of $\mathbb{E}[f(\widehat{w}) - \min_{w \in \mathcal{W}} f(w)] = O(D(G + \sigma)/\sqrt{T})$ (Lan, 2020, Theorem 4.1).

(iii) *Accelerated stochastic gradient descent* (accelerated SGD, Lan, 2020), which guarantees in the convex smooth setting that (Lan, 2020, Proposition 4.4)

$$\mathbb{E}[f(\widehat{w}) - f(w^\star)] = O(\beta D^2/T^2 + \sigma D/\sqrt{T}).$$

## 3. Asynchronous Mini-Batching

In this section we present and analyze a general template for mini-batching with asynchronous computation. The template, presented in Algorithm 1, receives a batch size parameter and a stochastic optimization algorithm $\mathcal{A}(\sigma, K)$, an algorithm performing $K$ point queries for stochastic gradients which has a variance bounded by $\sigma^2$ to compute an output point. Then the template uses the asynchronous computations to simulate mini-batched optimization, ignoring stale gradients with respect to the inner iterations of $\mathcal{A}$ according to their delays.

Following is our main result about Algorithm 1, which provides a black-box conversion of standard optimization rates

any $\tau_q \geq 0$ that satisfies both $\Pr(d \leq \tau_q) \geq q$ and $\Pr(d \geq \tau_q) \geq 1 - q$, where $d$ is sampled uniformly from $\{d_1, \ldots, d_T\}$. We also define the 1-quantile $\tau_1 \triangleq \tau_{\max}$.

to asynchronous optimization rates using the asynchronous mini-batching scheme.

---

**Algorithm 1:** Asynchronous mini-batching

**Input:** Batch size $B$, stochastic optimization algorithm $\mathcal{A}(\sigma, K)$

$t \leftarrow 1$       # *rounds count*
**for** $k \leftarrow 1, 2, \ldots, K$ **do**    # *queries count*
  Get query $\tilde{w}_k$ from $\mathcal{A}$
  $t_k \leftarrow t$      # *first step with $w_t = \tilde{w}_k$*
  $b \leftarrow 0$
  $\tilde{g}_k \leftarrow 0$
  **while** $b < B$ **do**
    Play $w_t = \tilde{w}_k$
    Receive $(g_t, d_t)$
    **if** $t_k \leq t - d_t$ **then**    # *ensures $w_{t-d_t} = \tilde{w}_k$*
      $\tilde{g}_k \leftarrow \tilde{g}_k + \frac{1}{B}g_t$
      $b \leftarrow b + 1$
    $t \leftarrow t + 1$
  Send response $\tilde{g}_k$ to $\mathcal{A}$
Output result produced by $\mathcal{A}$

---

**Theorem 1.** *Let $\mathcal{F}$ be some function class of differentiable functions, let $f \in \mathcal{F}$ and let $g$ be a stochastic first-order oracle with $\sigma^2$-bounded variance. Let $\mathcal{A}(\sigma, K)$ be a $K$-query stochastic first-order optimization algorithm for class $\mathcal{F}$ with a convergence rate guarantee of $Rate(f, \sigma, K)$. For $T$ asynchronous rounds, let $\tau_q$ be the $q$-quantile delay for some $q \in (0, 1]$, and let $B = \max\{1, \bar{\tau}_q\}$ for some $\bar{\tau}_q \geq \tau_q$. Then the output of Algorithm 1 with batch size $B$ and $\mathcal{A}(\sigma/\sqrt{B}, \lfloor qT/(1 + 2\bar{\tau}_q) \rfloor)$ has a convergence rate guarantee of $Rate(f, \sigma/\sqrt{\max\{1, \bar{\tau}_q\}}, \lfloor qT/(1 + 2\bar{\tau}_q) \rfloor)$.*

Theorem 1 uses a delay bound of the $q$-quantile delay and an optimization algorithm, inheriting the convergence rate of the original algorithm with a modified effective number of steps that depends on the delay quantile and decreased stochastic variance bound due to the mini-batching. Note that the above guarantee can be significantly stronger than that of naive synchronous mini-batching, where the iteration time is determined by the slowest machine, since the proposed method allows faster machines to contribute multiple times per batch. To better understand the result we present the following corollary, which is a direct application of the theorem with the classical stochastic methods detailed in Appendix A.

**Corollary 2.** *Let $f : \mathcal{W} \to \mathbb{R}$ be a differentiable function, $g : \mathcal{W} \to \mathbb{R}^d$ a first-order oracle of $f$ with variance bounded by $\sigma^2 \geq 0$, $T > 0$ the number of asynchronous rounds, $q \in (0, 1]$ and $\bar{\tau}_q > 0$ such that $\tau_q \leq \bar{\tau}_q$, where $\tau_q$ is the $q$-quantile delay. Then the following holds:*

(i) *In the non-convex smooth setting, Algorithm 1 with tuned SGD and batch size of $\max\{1, \bar{\tau}_q\}$ produce $\widehat{w}$*

*which satisfy*

$$\mathbb{E}[\|\nabla f(\widehat{w})\|^2] = O\left(\frac{(1 + \bar\tau_q)\beta F}{qT} + \frac{\sigma\sqrt{\beta F}}{\sqrt{qT}}\right).$$

*(ii) In the convex smooth setting, Algorithm 1 with tuned SGD and batch size of $\max\{1, \bar\tau_q\}$ produce $\widehat{w}$ which satisfy*

$$\mathbb{E}[f(\widehat{w}) - f(w^\star)] = O\left(\frac{(1 + \bar\tau_q)\beta D^2}{qT} + \frac{D\sigma}{\sqrt{qT}}\right).$$

*(iii) In the convex smooth setting, Algorithm 1 with tuned accelerated SGD and batch size of $\max\{1, \bar\tau_q\}$ produce $\widehat{w}$ which satisfy*

$$\mathbb{E}[f(\widehat{w}) - f(w^\star)] = O\left(\frac{(1 + \bar\tau_q)^2\beta D^2}{(qT)^2} + \frac{D\sigma}{\sqrt{qT}}\right).$$

*(iv) In the convex Lipschitz setting, Algorithm 1 with tuned projected SGD and batch size of $\max\{1, \bar\tau_q\}$ produce $\widehat{w}$ which satisfy*

$$\mathbb{E}[f(\widehat{w}) - \min_{w \in \mathcal{W}} f(w)] = O\left(\frac{D(\sqrt{1 + \bar\tau_q}G + \sigma)}{\sqrt{qT}}\right).$$

*Proof.* The result follows by a simple application of Theorem 1, which provides a rate of

$$\text{Rate}(f, \sigma/\sqrt{\max\{1, \bar\tau_q\}}, \lfloor qT/(1 + 2\bar\tau_q) \rfloor),$$

with each of the standard convergence guarantees (Lemmas 3 to 6 in Appendix A).  □

We shortly remark that, as the median delay is bounded by twice the average delay and may be much smaller, and as the number of machines is always larger than the average delay (see Appendices B and C for details), our results provide tighter guarantees than existing results that depend on an average delay bound or the number of machines (Cohen et al., 2021; Mishchenko et al., 2022; Koloskova et al., 2022; Feyzmahdavian & Johansson, 2023), without requiring a specialized optimization analysis in the asynchronous setting. In addition, we are the first to provide an accelerated rate in the convex smooth setting.

Following is the key lemma for proving Theorem 1, which provides a condition to ensure that $\mathcal{A}$ receives sufficiently many updates so as to produce a effective result. Its proof follows.

**Lemma 1.** *Let $B \in \mathbb{N}$ and $\mathcal{A}(\sigma, K)$ a stochastic optimization algorithm for some $K \in \mathbb{N}$ and $\sigma \geq 0$. Then when running Algorithm 1 with batch size $B$ and algorithm $\mathcal{A}$ for*

*$T$ asynchronous rounds, a sufficient condition for an output to be produced is*

$$K \leq \sup_{q \in (0,1]} \left\lfloor \frac{qT}{B + \tau_q} \right\rfloor.$$

*Proof of Lemma 1.* Let $K'$ be the number of responses $\mathcal{A}$ receives and assume by contradiction that $K' < K$. Let $q \in (0, 1]$. At least $\lceil qT \rceil$ rounds have a delay less or equal $\tau_q$. Let $n(k)$ be the number of asynchronous rounds with delay less or equal $\tau_q$ at times $t_k \leq t < t_{k+1}$. Assume by contradiction that $n(k) > B + \tau_q$ for some $k$, and let $S = (a_1, \ldots, a_{n(k)})$ be the rounds during the $k$'th iteration with $d_{a_i} \leq \tau_q$ ordered in an increasing order. For any $i > \tau_q$ (which implies that $i \geq \tau_q + 1$ as both are integers), as the sequence is non-decreasing and $a_1 \geq t_k$,

$$a_i - t_k \geq a_i - a_1 \geq i - 1 \geq \tau_q \geq d_{a_i},$$

which means that the inner "if" condition is satisfied and contradicts the condition of the while loop, $b \leq B$, since there are at least $B + 1$ rounds $a_i \in S$ with $i > \tau_q$ (because our assumption by contradiction yields $|S| \geq B + \tau_q + 1$ as we deal with integers). Thus, $n(k) \leq B + \tau_q$ for all $k \in [K']$. As each $B + \tau_q$ consecutive rounds with delay less or equal $\tau_q$ must account for a response to $\mathcal{A}$, and we have at least $\lceil qT \rceil$ such rounds,

$$K' \geq \left\lfloor \frac{qT}{B + \tau_q} \right\rfloor.$$

As this is true for any $q \in (0, 1]$,

$$K' \geq \sup_{q \in (0,1]} \left\lfloor \frac{qT}{B + \tau_q} \right\rfloor \geq K$$

which contradicts the assumption $K' < K$.  □

We proceed to prove the main result.

*Proof of Theorem 1.* By Lemma 1 $\widehat{w}$ is produced since

$$\sup_{q' \in (0,1]} \left\lfloor \frac{q'T}{B + \tau_{q'}} \right\rfloor \geq \left\lfloor \frac{qT}{B + \tau_q} \right\rfloor \geq \left\lfloor \frac{qT}{1 + 2\bar\tau_q} \right\rfloor = K.$$

The $t_k \leq t - d_t$ condition ensures that the gradient at round $t$ is of $\tilde{w}_k$, as $w_t = \tilde{w}_k$ for all $t_k \leq t < t_{k+1}$. We conclude using the linearity of the variance to bound the variance of the mini-batch,

$$\mathbb{E}[\|\tilde{g}_k - \nabla f(\tilde{w}_k)\|^2] \leq \frac{\sigma^2}{\max\{1, \bar\tau_q\}},$$

invoking the rate guarantee of

$$\mathcal{A}(\sigma/\sqrt{\max\{1, \bar\tau_q\}}, \lfloor qT/(1 + 2\bar\tau_q) \rfloor).  □$$

## 4. Quantile Adaptivity

The method described in the previous section require as input an upper bound of a given quantile of the delays. Next, we show how to achieve more robust guarantees without prior knowledge about the delays, by establishing guarantees that are competitive with respect to all quantiles simultaneously.

First, let us clarify our objective. We begin with the smooth non-convex case. Let $q \in (0, 1]$ be an arbitrary quantile. Using Corollary 2 with the tight quantile delay bound $\tau_q$, we can obtain a rate of

$$O\left(\frac{(1 + \tau_q)\beta F}{qT} + \frac{\sigma\sqrt{\beta F}}{\sqrt{qT}}\right).$$

As each $q \in (0, 1]$ produce a difference guarantee, our objective is to provide a guarantee of

$$O\left(\inf_{q \in (0,1]} \frac{(1 + \tau_q)\beta F}{qT} + \frac{\sigma\sqrt{\beta F}}{\sqrt{qT}}\right),$$

without knowing in advance which quantile $q$ achieves this infimum.

Our quantile-adaptive template of asynchronous mini-batching appears in Algorithm 2. Since the best quantile (for which the best convergence rate is achieved) is not known in advance, it cannot be used to tune the batch size. For that reason, the template extends Algorithm 1 by performing multiple asynchronous mini-batching iterations with different batch sizes and time horizons, essentially making it into an any-time algorithm using a doubling procedure (Cesa-Bianchi & Lugosi, 2006). By carefully selecting the parameters, we can achieve the desired quantile adaptivity.

Following is our second main result which provides quantile adaptivity by coupling Algorithm 2 with SGD and accelerated SGD. Additional results using SGD in the convex and convex smooth settings appear in Appendix E.

**Theorem 3.** *Let $\mathcal{W} \subseteq \mathbb{R}^d$ be a convex set, $f : \mathcal{W} \to \mathbb{R}$ be a differentiable function, $g : \mathcal{W} \to \mathbb{R}^d$ an unbiased gradient oracle of $f$ with $\sigma^2$-bounded variance, $T \in \mathbb{N}$ and $w_1 \in \mathcal{W}$. Running Algorithm 2 for $T$ asynchronous steps with some parameters $\mathcal{A}$, $B_1, B_2, \ldots$ and $T_1, T_2, \ldots$, let $W = (\widehat{w}_1, \ldots, \widehat{w}_I)$ be the outputs of $\mathcal{A}$, and let $\widehat{w}$ be $\widehat{w}_I$ if $W$ is non-empty and $w_1$ otherwise. Then the following holds:*

*(i) If $\mathcal{W} = \mathbb{R}^d$, $f$ is $\beta$-smooth and lower bounded by $f^\star$, setting $\mathcal{A}(K)$ to be $K$-steps SGD initialized at $w_1$ with stepsize $\frac{1}{\beta}$,*

$$K_i = 2^{i-1} \quad and \quad B_i = \max\left\{1, \left\lceil \frac{\sigma^2 K_i}{2\beta F} \right\rceil\right\},$$

---

**Algorithm 2:** Asynchronous mini-batching sweep

**Input:** Batch sizes $B_i$, Epoch lengths $K_i$, $K$-query algorithm $\mathcal{A}(K)$[4]

$t \leftarrow 1$           # rounds count
**for** $i \leftarrow 1, 2, \ldots$ **do**
     Initialize $\mathcal{A}(K_i)$
     **for** $k \leftarrow 1, 2, \ldots, K_i$ **do** # query count at $i^{th}$ epoch
         Get query $\tilde{w}_k^{(i)}$ from $\mathcal{A}$
         $t_k^{(i)} \leftarrow t$        # first step with $w_t = \tilde{w}_k^{(i)}$
         $b \leftarrow 0$
         $\tilde{g}_k^{(i)} \leftarrow 0$
         **while** $b < B_i$ **do**
             Play $w_t = \tilde{w}_k^{(i)}$
             Receive $(g_t, d_t)$
             **if** $t_k^{(i)} \leq t - d_t$ **then** # ensures $w_{t-d_t} = \tilde{w}_k^{(i)}$
                 $\tilde{g}_k^{(i)} \leftarrow \tilde{g}_k^{(i)} + \frac{1}{B_i} g_t$
                 $b \leftarrow b + 1$
             $t \leftarrow t + 1$
         Send response $\tilde{g}_k^{(i)}$ to $\mathcal{A}$
     Receive output $\widehat{w}_i$ from $\mathcal{A}$

---

*for some $F \geq f(w_1) - f^\star$, then*

$$\mathbb{E}\left[\|\nabla f(\widehat{w})\|^2\right] \leq \inf_{q \in (0,1]} \frac{24(1 + 2\tau_q)\beta F}{qT} + \frac{24\sigma\sqrt{\beta F}}{\sqrt{qT}}.$$

*(ii) If $\mathcal{W} = \mathbb{R}^d$, $f$ is convex, $\beta$-smooth and admits a minimizer $w^\star$, setting $\mathcal{A}(K)$ to be $K$-steps accelerated SGD initialized at $w_1$ with stepsize $\frac{1}{4\beta}$, $K_i = 2^{i-1}$, and*

$$B_i = \max\left\{1, \left\lceil \frac{\sigma^2 K_i (K_i + 1)^2}{12\beta^2 D^2} \right\rceil\right\}.$$

*for some $D \geq \|w_1 - w^\star\|$, then $\mathbb{E}[f(\widehat{w}) - f(w^\star)]$ is bounded by*

$$\inf_{q \in (0,1]} \frac{192(1 + 2\tau_q)^2\beta D^2}{q^2 T^2} + \frac{72\sigma D}{\sqrt{qT}}.$$

Note that the infimum over quantiles in the statement above is at least as good as the median (or any other given quantile), improving upon Corollary 2 by automatically adapting to the best quantile without requiring any prior knowledge of a quantile upper bound. This is achieved by maximizing the batch size according to the underlying rate of $\mathcal{A}$ instead of using the delay bound. As discussed at length in Sections 1.1 and 1.2, this adaptivity to quantile delays is more robust than existing average delay dependent results, and can further achieve accelerated guarantees for convex smooth objectives.

---

[4]We omit the variance input parameter of $\mathcal{A}$ in Algorithm 2 since it is not used.

Next is the key lemma of the procedure which links the total number of rounds $T$ with the effective number of responses $\mathcal{A}$ receives. It is essentially an extension of Lemma 1 and follows the same proof technique. For the proof see Appendix F.

**Lemma 2.** *Let $B_1, B_2, \ldots$ be a non-decreasing sequence of positive integers and $K_i = 2^{i-1}$ for $i \in \mathbb{N}$. Running Algorithm 2 for $T \in \mathbb{N}$ asynchronous steps, let $W = (\widehat{w}_1, \ldots, \widehat{w}_I)$ be the outputs of $\mathcal{A}$. Then assuming $W$ is not empty, for any $q \in (0, 1]$,*

$$qT < 2(B_{I+1} + \tau_q)K_{I+1}.$$

Following is the proof of item (i) of Theorem 3; we defer the proof of other items to Appendix F.

*Proof of Theorem 3 (i).* If $W$ is empty, then $T < B_1$ (as $T_1 = 1$ and the condition $1 = t_1^{(1)} \leq t - d_t$ is always satisfied). Hence, as $T \geq 1$, $B_1 > 1$ and $T < \frac{\sigma^2}{2\beta F}$, and from smoothness,

$$\|\nabla f(\widehat{w})\|^2 = \|\nabla f(w_1)\|^2 \leq 2\beta F < \frac{\sigma^2}{T} \leq \frac{\sigma^2}{qT}$$

for all $q \in (0, 1]$, where the first inequality follows by using a standard property of smooth functions, $\|\nabla f(w)\|^2 \leq 2\beta(f(w) - f^\star)$, which is established by the smoothness property between $w^+ = w - \frac{1}{\beta}\nabla f(w)$ and $w$,

$$f^\star - f(w) \leq f(w^+) - f(w)$$
$$\leq \nabla f(w) \cdot (w^+ - w) + \frac{\beta}{2}\|w^+ - w\|^2$$
$$= -\frac{1}{2\beta}\|\nabla f(w)\|^2.$$

We proceed to the case where $W$ is not empty. Let $q \in (0, 1]$. By Lemma 2, $qT < 2(B_{I+1} + \tau_q)K_{I+1}$. If $B_{I+1} \leq \max\{1, \tau_q\}$,

$$qT \leq 2(1 + 2\tau_q)K_{I+1} \implies \frac{1}{K_I} \leq \frac{4(1 + 2\tau_q)}{qT}.$$

If $B_{I+1} > \max\{1, \tau_q\}$,

$$qT \leq 4B_{I+1}K_{I+1} \leq \frac{4\sigma^2 K_{I+1}^2}{\beta F} \implies \frac{1}{K_I} \leq \frac{4\sigma}{\sqrt{\beta F qT}}.$$

We are left with applying a standard convergence result for SGD, which is detailed in Lemma 3 and establishes that

$$\mathbb{E}\left[\|\nabla f(\widehat{w})\|^2\right] \leq \frac{2\beta F}{K} + \sqrt{\frac{8\sigma^2\beta F}{K}},$$

where $\widehat{w}$ is the output (a uniformly sampled iterate) of $K$-steps SGD with a fixed stepsize $\gamma = \min\{1/\beta, \sqrt{2F/\sigma^2\beta K}\}$.

To that end note that $\tilde{g}_k^{(I)}$ is the average of $B_I$ i.i.d. unbiased estimations of $\nabla f(\tilde{w}_k^{(I)})$, and by the linearity of the variance has variance bounded by $\sigma^2/B_I$. Thus, our selection of parameters enables us to use Lemma 3 and establish that

$$\mathbb{E}\left[\|\nabla f(w_I)\|^2\right] \leq \frac{2\beta F}{K_I} + \sqrt{\frac{8\sigma^2\beta F}{B_I K_I}} \leq \frac{6\beta F}{K_I}$$
$$\leq \frac{24(1 + 2\tau_q)\beta F}{qT} + \frac{24\sigma\sqrt{\beta F}}{\sqrt{qT}}.$$

As this holds for any $q \in (0, 1]$,

$$\mathbb{E}\left[\|\nabla f(w_I)\|^2\right] \leq \inf_{q \in (0,1]} \frac{24(1 + 2\tau_q)\beta F}{qT} + \frac{24\sigma\sqrt{\beta F}}{\sqrt{qT}}. \quad \square$$

## 5. Experimental Evaluation

To illustrate the benefits of asynchronous mini-batching, we compare "vanilla" asynchronous SGD (denoted Async-SGD) with a practical variant of our mini-batch method (Algorithm 1), which uses SGD, denoted Async-MB-SGD, for training a fully connected neural network on the Fashion-MNIST classification dataset (Xiao et al., 2017).[5] The dataset consists of 60,000 training images and 10,000 test images, each of size $28 \times 28$ pixels and labeled across 10 classes. We use test accuracy as the evaluation metric.

**Practical modifications.** Consider a scenario with a constant delay, $d_t = \tau$, for all $t \in [T]$. When running Algorithm 1 with batch size $\tau$, only one model update occurs every $2\tau - 1$ gradient computations, and $\tau - 1$ of those gradients are discarded. Although the large amount of discarded gradients only affects the constant factors in the theoretical analysis, it is inefficient in practice. To avoid discarding many gradient computations that are only slightly stale, we modify the condition "**if** $t_k \leq t - d_t$ **then**" in Algorithm 1 to "**if** $t_{\max\{k-2,1\}} \leq t - d_t$ **then**". This change allows the use of slightly older gradients, resulting in an effective delay of at most 2, while increasing the proportion of computations that contribute to model updates. Although this modification introduces a small amount of gradient staleness, the improved utilization of computed gradients is beneficial in practice. Our experiments report performance using this variant.

**Asynchronous workers setup.** We adopt the two-phase asynchronous simulation framework of Cohen et al. (2021). In the first phase, we simulate compute times for each worker by drawing from a weighted mixture of two Poisson distributions. In the second phase, we simulate training by having each worker deliver gradients to a central server

---

[5]We also experimented with filtering gradients whose delays exceed the number of machines, as proposed by Koloskova et al. (2022), but observed performance similar to that of Async-SGD.

Table 2: Fashion-MNIST test accuracies for Async-SGD and Async-MB-SGD across different numbers of workers and update steps. The test accuracy column reports mean ± standard deviation across 3 runs with different seeds.

| NUM. WORKERS | NUM. STEPS | METHOD | LEARNING RATE | TEST ACCURACY |
|---|---|---|---|---|
| 40 | 7,500 | Async-SGD | 0.021544 | $0.8449 \pm 0.0013$ |
| | | Async-MB-SGD($B=2$) | 0.100000 | $\mathbf{0.8457} \pm 0.0004$ |
| 160 | 30,000 | Async-SGD | 0.010000 | $0.8585 \pm 0.0017$ |
| | | Async-MB-SGD($B=8$) | 0.215443 | $\mathbf{0.8620} \pm 0.0008$ |
| 640 | 120,000 | Async-SGD | 0.002154 | $0.8623 \pm 0.0015$ |
| | | Async-MB-SGD($B=8$) | 0.100000 | $\mathbf{0.8737} \pm 0.0002$ |

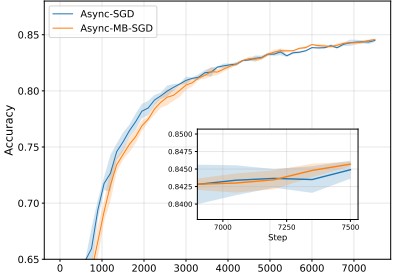 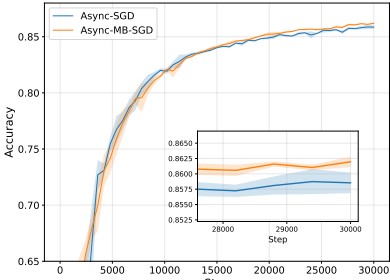 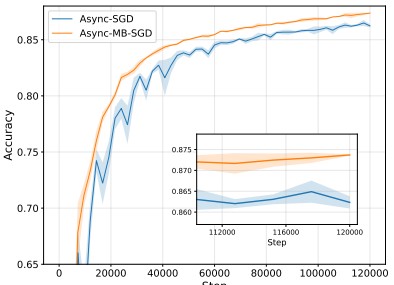

Figure 1: Comparison of Fashion-MNIST test accuracy for Async-SGD and Async-MB-SGD with varying numbers of workers and update steps. (left) 40 workers, 7,500 steps, (middle) 160 workers, 30,000 steps, (right) 640 workers, 120,000 steps. Each plot presents the mean and standard deviation across 3 runs.

according to the generated compute schedule. Specifically, we follow schedule B of Cohen et al. (2021), where each compute time is drawn from a Poisson distribution with parameter $P$ with probability 0.92 and from a Poisson distribution with parameter $150P$ with probability 0.08, using $P = 4.06$. To avoid compute times of 0, the Poisson distributions are shifted by +1. In contrast to Cohen et al. (2021), we sample a single compute time per update instead of separating the gradient computation and the gradient update. We conduct experiments with 40, 160, and 640 workers, using 7,500, 30,000, and 120,000 update steps, respectively. The linear relationship between the number of workers and update steps reflects the increase in total computation when scaling the number of workers over a fixed time budget.

**Model and training setup.** We train a three-layer fully connected neural network with input dimension $728 = 28^2$, hidden layers of sizes 256 and 128, an output layer of size 10, and ReLU activations. The model is trained using the cross-entropy loss. To reduce the variation of the last iterate, we use exponential moving averaging with decay 0.99. Each worker uses a local mini-batch of size 8. The learning rate is selected separately for each algorithm from a geometric grid with multiplicative factor $\sqrt[3]{10}$: for Async-SGD we search over the range [0.001, 1.0], and for Async-MB-SGD over [0.01, 1.0]. For Async-MB-SGD, we additionally tune the aggregation batch size $B$ (i.e., the number of updates the server accumulates before modifying the model) over the

set $\{1, 2, 4, 8, 16, 32\}$. For the best set of hyperparameters, we report the mean and standard deviation across 3 runs.

## 5.1. Results and Discussion

Table 2 reports test accuracy under various worker–step configurations. Async-SGD and Async-MB-SGD perform similarly with 40 workers, with a slight edge to Async-MB-SGD with an accuracy of 84.57%. On the other hand, when the number of workers increases to 160 and 640, Async-MB-SGD consistently outperforms Async-SGD, with accuracy gaps of 0.35% and 1.1%. Notably, Async-MB-SGD uses a significantly larger stepsize, which aligns with expectations: the method experiences smaller effective delays due to gradient filtering and benefits from reduced variance via mini-batching. Figure 1 presents test accuracy plots under different configurations. We observe smoother curves with Async-MB-SGD, particularly at larger worker counts. This behavior likely also results from the method's ability to reduce the effective delay of gradient updates.

These results highlight the robustness of Async-MB-SGD to the scale of the number of workers. By mitigating the effects of delayed updates through gradient filtering and mini-batching, the method maintains strong performance and supports larger learning rates even under highly asynchronous conditions. This suggests that Async-MB-SGD can serve as a practical alternative to vanilla asynchronous SGD in large-scale distributed training settings.

## Acknowledgements

This project has received funding from the European Research Council (ERC) under the European Union's Horizon 2020 research and innovation program (grant agreement No. 101078075). Views and opinions expressed are however those of the author(s) only and do not necessarily reflect those of the European Union or the European Research Council. Neither the European Union nor the granting authority can be held responsible for them. This work received additional support from the Israel Science Foundation (ISF, grant numbers 2549/19 and 3174/23), a grant from the Tel Aviv University Center for AI and Data Science (TAD), from the Len Blavatnik and the Blavatnik Family foundation, from the Prof. Amnon Shashua and Mrs. Anat Ramaty Shashua Foundation, and a fellowship from the Israeli Council for Higher Education.

## Impact Statement

This paper presents work whose goal is to advance the field of Machine Learning. There are many potential societal consequences of our work, none which we feel must be specifically highlighted here.

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

## A. Standard Convergence Bounds in Stochastic Optimization

Convergence result of SGD for non-convex smooth optimization from Lan (2020) (Corollary 6.1).

**Lemma 3.** *Let $f : \mathbb{R}^d \to \mathbb{R}$ be a $\beta$-smooth function lower bounded by $f^\star$. Let $g : \mathbb{R}^d \to \mathbb{R}^d$ be an unbiased gradient oracle of $f$ with $\sigma^2$-bounded variance. Then running SGD (termed RSGD in Lan, 2020) for $K$ steps, initialized at $w_1 \in \mathbb{R}^d$ and with stepsizes $\gamma_k = \min\{1/\beta, \sqrt{2F/\sigma^2\beta K}\}$, where $F \geq f(w_1) - f^\star$, produce $\widehat{w}$ which satisfy*

$$\mathbb{E}\left[\|\nabla f(\widehat{w})\|^2\right] \leq \frac{2\beta F}{K} + \sqrt{\frac{8\sigma^2\beta F}{K}}.$$

Convergence result of SGD for convex smooth optimization from Lan (2020) (Corollary 6.1).

**Lemma 4.** *Let $f : \mathbb{R}^d \to \mathbb{R}$ be a $\beta$-smooth convex function admitting a minimizer $w^\star$. Let $g : \mathbb{R}^d \to \mathbb{R}^d$ be an unbiased gradient oracle of $f$ with $\sigma^2$ bounded variance. Then running SGD (termed RSGD in Lan, 2020) for $K$ steps, initialized at $w_1 \in \mathbb{R}^d$ and with stepsizes $\gamma_k = \min\{1/\beta, \sqrt{D^2/\sigma^2 K}\}$, where $D \geq \|w_1 - w^\star\|$, produce $\widehat{w}$ which satisfy*

$$\mathbb{E}\left[f(\widehat{w}) - f(w^\star)\right] \leq \frac{\beta D^2}{K} + \frac{2\sigma D}{\sqrt{K}}.$$

Convergence result of accelerated SGD (AC-SA) for convex smooth optimization from Lan (2020) (Proposition 4.4).

**Lemma 5.** *Let $f : \mathbb{R}^d \to \mathbb{R}$ be a $\beta$-smooth convex function admitting a minimizer $w^\star$. Let $g : \mathbb{R}^d \to \mathbb{R}^d$ be an unbiased gradient oracle of $f$ with $\sigma^2$ bounded variance. Then running accelerated SGD (termed AC-SA in Lan, 2020) for $K$ steps, initialized at $w_1 \in \mathbb{R}^d$ and with parameters $\alpha_t = \frac{2}{t+1}$ and $\gamma_t = \gamma t$ where*

$$\gamma = \min\left\{\frac{1}{4\beta}, \sqrt{\frac{3D^2}{4\sigma^2 K(K+1)^2}}\right\}$$

*and $D \geq \|w_1 - w^\star\|$, produce $\widehat{w}$ which satisfy*

$$\mathbb{E}\left[f(\widehat{w}) - f(w^\star)\right] \leq \frac{4\beta D^2}{K(K+1)} + \frac{4\sigma D}{\sqrt{3K}}.$$

Convergence result of SGD for convex Lipschitz optimization from Lan (2020) (Theorem 4.1, Equation 4.1.12).

**Lemma 6.** *Let $\mathcal{W} \subset \mathbb{R}^d$ be a convex set with diameter bounded by $D$. Let $f : \mathcal{W} \to \mathbb{R}$ be a $G$-Lipschitz convex function and $w^\star \in \arg\min_{w \in \mathcal{W}} f(w)$. Let $g : \mathbb{R}^d \to \mathbb{R}^d$ be an unbiased sub-gradient oracle of $f$ with $\sigma^2$ bounded variance. Then running projected SGD (termed stochastic mirror descent in Lan, 2020) for $K$ steps, initialized at $w_1 \in \mathcal{W}$ and with stepsizes $\gamma_t = D/\sqrt{(G^2 + \sigma^2)K}$, produce $\widehat{w}$ which satisfy*

$$\mathbb{E}\left[f(\widehat{w}) - f(w^\star)\right] \leq \frac{2D\sqrt{G^2 + \sigma^2}}{\sqrt{K}}.$$

## B. Average Delay is Bounded by Number of Machines

In the arbitrary delay model, it was previously remarked (Koloskova et al., 2022; Feyzmahdavian & Johansson, 2023) that if the gradients are produced by a constant number of machines (the setting Koloskova et al. (2022); Mishchenko et al. (2022) considered), the average delay is smaller than the number of machines. For completeness we state and prove this property below.

**Lemma 7.** *Let $d_1, \ldots, d_T$ be an arbitrary delay sequence produced by $M$ machines. Then the average delay is lower bounded by the number of machines. In particular, $\frac{1}{T}\sum_{t=1}^{T} d_t \leq M - 1$.*

*Proof.* For any $m \in [M]$, let $S_m = \{k_{m,1}, k_{m,2}, \ldots, k_{m,n(m)}\}$ be the rounds where the gradient was computed by machine $m$, where $n(k)$ is the total number of gradients produced by machine $m$. Rearranging the summation of delays and treating

$k_{m,0} = 0$,

$$\sum_{t=1}^{T} d_t = \sum_{m \in [M]} \sum_{i=1}^{n(m)} k_{m,i} - k_{m,i-1} - 1 = -T + \sum_{m \in [M]} \sum_{i=1}^{n(m)} k_{m,i} - k_{m,i-1}$$

$$= -T + \sum_{m \in [M]} k_{m,n(m)}.$$

As $k_{m,n(m)} \neq k_{m',n(m')}$ if $m \neq m'$ (as each step has only a single gradient),

$$\sum_{t=1}^{T} d_t = -T + \sum_{m \in [M]} k_{m,n(m)} \leq -T + \sum_{t=T-M+1}^{T} t = -T + \frac{M(T-M+1+T)}{2}$$

$$= \frac{2MT - M^2 + M - 2T}{2} \leq T(M-1).$$

Dividing both sides by $T$ we conclude $\frac{1}{T} \sum_{t=1}^{T} d_t \leq M - 1$. $\qquad\square$

## C. Best Quantile Bound Arbitrarily Improves over Average and Median

As we establish in Theorem 1 and Corollary 2, given a bound of the median delay (or any other quantile), it is possible to convert classical optimization methods to asynchronous optimization methods in a black-box manner. Note that by a simple application of Markov's inequality, the median delay is bounded by twice the average delay, as $\tau_{\text{med}} \leq \tau_{\text{avg}}/\Pr(d_t \geq \tau_{\text{med}}) \leq 2\tau_{\text{avg}}$, where $t$ is chosen uniformly at random. On the other hand, for the delay sequence $d_t = \mathbf{1}[t > T/2 + 1](t-1)$, we have $\tau_{\text{avg}} = \Omega(T)$ and a significantly smaller $\tau_{\text{med}} = 0$.

A natural question is whether we can improve beyond the average or median delay dependency. To gain intuition consider the following simple delay sequence, which is produced by performing asynchronous optimization with $M$ machines where one machine is $n$ times as fast for $T = n + M - 1$ steps,

$$d_t = \begin{cases} 0 & \text{if } t \leq n; \\ t - 1 & \text{otherwise.} \end{cases}$$

In the $\beta$-smooth non-convex case, assuming $\sigma = 0$, it is straightforward to see that the best approach is to perform SGD and ignore all stale gradients, as they do not provide additional information, achieving a convergence rate of

$$O\left(\frac{\beta(f(w_1) - f^\star)}{n}\right) = O\left(\frac{\beta(f(w_1) - f^\star)}{T \cdot \frac{n}{n+M-1}}\right) = O\left(\frac{\beta(f(w_1) - f^\star)}{qT}\right),$$

where $q = \frac{n+M-1}{n}$ is the maximal quantile of the gradients with zero delay. In case $q < \frac{1}{2}$, the median and average delays will be $\Omega(n)$, and dependence on them will lead to far worse guarantees. To that end, we would like a result more robust to the delay distribution which better ignores outliers.

## D. Lower Bound for "Vanilla" Fixed Stepsize Asynchronous SGD

The following theorem shows that, without further assumptions or modifications to the algorithm, the performance of vanilla (non-adaptive) asynchronous SGD with a fixed stepsize must degrade according to the *maximal delay*, similarly to the upper bound of Stich & Karimireddy (2020). In particular, assuming $\tau_{\text{max}} = o(T)$, the delay sequence defined in Theorem 4 satisfies $\tau_{\text{avg}} = o(\tau_{\text{max}})$. The result may seem inconsistent with the upper bound of asynchronous SGD from Koloskova et al. (2022), which degrades as $\sqrt{\tau_{\text{avg}} \tau_{\text{max}}}$. However, this is not the case, as their definition of average delay includes "imaginary" delays from machines that return results after round $T$, rather than accounting solely for the observed sequence.

**Theorem 4.** *For any $T \in \mathbb{N}$, $\tau_{max} \in [T-2]$ and $w_1 \in \mathbb{R}$, there exists a delay sequence of length $T$ with maximal delay $\tau_{max}$ and average delay bounded by $\tau_{max}^2/T$, which can be produced by $\tau_{max} + 1$ machines, and a $\beta$-smooth convex function $f : \mathbb{R} \to \mathbb{R}$ admitting a minimizer $w^\star$, such that for any $\eta > \frac{6}{\beta(1+\tau_{max})}$, the iterates of $T$-steps asynchronous SGD with the delay sequence and deterministic gradients, initialized at $w_1$ with stepsize $\eta$, satisfy*

$$\frac{1}{T} \sum_{t=1}^{T} \|\nabla f(w_t)\|^2 \geq \frac{4(1+\tau_{max})\beta(f(w_1) - f(w^\star))}{T}$$

*and*

$$\frac{1}{T}\sum_{t=1}^{T}f(w_t) - f(w^\star) \geq \frac{(1 + \tau_{max})\beta\|w_1 - w^\star\|^2}{T}.$$

Note that the above theorem holds for $\eta > 6/\beta(1 + \tau_{max})$. For a smaller value of $\eta$, the worst-case convergence guarantee of standard SGD (even without noise or delays) is $\Omega((1 + \tau_{max})\beta/T)$, which is also tight in the worst case. We cover this case in Appendix D.1.

*Proof of Theorem 4.* Let $f(w) = \frac{\beta}{2}\|w\|^2$ which is $\beta$-smooth and convex function admitting a minimizer $w^\star = 0$. We consider the following delay sequence,

$$d_t = \begin{cases} t - 1 & \text{if } t \leq \tau_{max} + 1; \\ 0 & \text{otherwise.} \end{cases}$$

Note that the maximal delay is $\tau_{max}$ and

$$\tau_{avg} = \frac{1}{T}\sum_{t=1}^{\tau_{max}+1}t - 1 = \frac{(\tau_{max} + 1)\tau_{max}}{2T} \leq \frac{\tau_{max}^2}{T}.$$

In addition, this sequence can be made with $\tau_{max} + 1$ machines, where the first $\tau_{max} + 1$ rounds are produced by different machines and the rest are produced by the last. The trajectory of the first $\tau_{max} + 2$ iterates can be written as

$$w_t = w_1 - \eta(t - 1)\nabla f(w_1) = w_1(1 - \eta\beta(t - 1)).$$

Thus, the average squared gradient norm can be bounded by

$$\frac{1}{T}\sum_{t=1}^{T}\|\nabla f(w_t)\|^2 \geq \frac{\beta^2}{T}\sum_{t=\tau_{max}/2+2}^{\tau_{max}+2}(1 - \eta\beta(t - 1))^2\|w_1\|^2.$$

Note that we treat $\tau_{max}$ as an even number, this is done for simplicity and the odd case affects only a slight constant modification as $\tau_{max} + 1 \leq 2\tau_{max}$. As $\eta > \frac{6}{\beta(\tau_{max}+1)}$, for $t \geq \tau_{max}/2 + 2$, $\eta\beta(t - 1) \geq 3$ and

$$\frac{1}{T}\sum_{t=1}^{T}\|\nabla f(w_t)\|^2 \geq \frac{\beta^2\|w_1\|^2}{T}\sum_{t=\tau_{max}/2+2}^{\tau_{max}+2}4$$

$$\geq \frac{2\beta^2\|w_1\|^2(\tau_{max} + 1)}{T}.$$

By a standard application of smoothness, $f(w_1) - f(w^\star) \leq \frac{\beta}{2}\|w_1 - w^\star\|^2$. Hence,

$$\frac{1}{T}\sum_{t=1}^{T}\|\nabla f(w_t)\|^2 \geq \frac{4\beta(f(w_1) - f(w^\star))(\tau_{max} + 1)}{T}.$$

And for the second inequality, as $f(w_t) - f(w^\star) = \frac{\beta}{2}\|w_t\|^2 = \frac{1}{2\beta}\|\nabla f(w_t)\|^2$,

$$\frac{1}{T}\sum_{t=1}^{T}f(w_t) - f^\star \geq \frac{1}{2\beta T}\sum_{t=1}^{T}\|\nabla f(w_t)\|^2$$

$$\geq \frac{\beta\|w_1 - w^\star\|^2(\tau_{max} + 1)}{T}. \qquad \square$$

### D.1. Lower Bound for "Vanilla" Asynchronous SGD with Small Fixed Stepsize

In Theorem 4 we provided a lower bound for $\beta$-smooth optimization using fixed stepsize asynchronous SGD with stepsize $\eta > \frac{6}{\beta(1+\tau_{\max})}$, where $\tau_{\max}$ is the maximal delay of the arbitrary delay sequence. Next, we complement this result with a simple lower bound for any delay sequence (which holds in particular for the centralized case by setting $d_t = 0$ for all $t \in [T]$), that has an inverse dependence on the stepsize, and yields for $\eta \leq \frac{6}{\beta(1+\tau_{\max})}$ the same lower bound of Theorem 4 (up to constant factors).

**Theorem 5.** *For any $T \in \mathbb{N}$, $w_1 \in \mathbb{R}$, $\beta > 0$ and $\eta > 0$, there exists a $\beta$-smooth convex function $f : \mathbb{R} \to \mathbb{R}$ admitting a minimizer $w^\star \in [-1, 1]$, such that for any delay sequence $d_1, \ldots, d_T$, the iterates of $T$-steps asynchronous (S)GD with the delay sequence and deterministic gradients, initialized at $w_1$ and with stepsize $\eta$, satisfy*

$$\frac{1}{T} \sum_{t=1}^{T} \|\nabla f(w_t)\|^2 \geq \frac{\beta(f(w_1) - f(w^\star))}{2 \max\{1, 2\beta\eta T\}}$$

*and*

$$\frac{1}{T} \sum_{t=1}^{T} f(w_t) - f(w^\star) \geq \frac{\beta \|w_1 - w^\star\|^2}{8 \max\{1, 2\beta\eta T\}}.$$

*Proof.* Without loss of generality, we will assume that $w_1 \geq 0$ and let $w^\star = -1$ (otherwise, use $w^\star = 1$ and a similar argument will hold). Let $f(w) = \frac{\epsilon}{2} \|w - w^\star\|^2$ for $\epsilon = \min\{\beta, 1/(2\eta T)\}$, which is $\beta$-smooth and convex function admitting a minimizer $w^\star$.

We will prove by induction that $\frac{1}{2}(w_1 - w^\star) \leq (w_t - w^\star) \leq (w_1 - w^\star)$ for all $t \in [T]$. The base case $t = 1$ is immediate since $w_1 - w^\star > 0$. At step $t$,

$$w_{t+1} = w_t - \eta \nabla f(w_{t-d_t}) = w_1 - \eta \sum_{s=1}^{t} \nabla f(w_{s-d_s}) = w_1 - \eta\epsilon \sum_{s=1}^{t} (w_{s-d_s} - w^\star).$$

Hence, by the induction assumption and the definition of $\epsilon$,

$$w_{t+1} - w^\star \geq (w_1 - w^\star) - \eta\epsilon t(w_1 - w^\star) \geq \frac{1}{2}(w_1 - w^\star).$$

On the other hand, as $w_{s-d_s} - w^\star \geq \frac{1}{2}(w_1 - w^\star) > 0$ by the induction,

$$(w_{t+1} - w^\star) = (w_1 - w^\star) - \eta\epsilon \sum_{s=1}^{t} (w_{s-d_s} - w^\star) \leq (w_1 - w^\star),$$

concluding the proof by induction. Hence,

$$\frac{1}{T} \sum_{t=1}^{T} \|\nabla f(w_t)\|^2 = \frac{\epsilon^2}{T} \sum_{t=1}^{T} \|w_t - w^\star\|^2 \geq \frac{\epsilon^2}{4T} \sum_{t=1}^{T} \|w_1 - w^\star\|^2 = \frac{\beta(f(w_1) - f(w^\star))}{2 \max\{1, 2\beta\eta T\}}$$

and

$$\frac{1}{T} \sum_{t=1}^{T} f(w_t) - f(w^\star) = \frac{\epsilon}{2T} \sum_{t=1}^{T} \|w_t - w^\star\|^2 \geq \frac{\epsilon}{8T} \sum_{t=1}^{T} \|w_1 - w^\star\|^2 = \frac{\beta \|w_1 - w^\star\|^2}{8 \max\{1, 2\beta\eta T\}}.$$

$\square$

## E. Additional Results using Algorithm 2

Following are additional applications of Algorithm 2 using projected SGD in the convex non-smooth and SGD in the convex smooth setting. Their proofs follow.

**Theorem 6.** *Let $\mathcal{W} \subset \mathbb{R}^d$ be a convex set with diameter bounded by $D$, $f : \mathcal{W} \to \mathbb{R}$ be a G-Lipschitz convex function and $w^\star \in \arg\min_{w \in \mathcal{W}} f(w)$. Let $g : \mathcal{W} \to \mathbb{R}^d$ be an unbiased gradient oracle of $f$ with $\sigma^2$-bounded variance, $T \in \mathbb{N}$ and $w_1 \in \mathcal{W}$. Let $\mathcal{A}(K)$ be stochastic mirror descent initialized at $w_1$ with stepsize*

$$\gamma_t = \frac{D}{\sqrt{(G^2 + \sigma^2/B)K}},$$

*where*

$$B = \max\left\{1, \left\lceil \frac{\sigma^2}{G^2} \right\rceil\right\}.$$

*Let $W = (\widehat{w}_1, \ldots, \widehat{w}_I)$ be the outputs of $\mathcal{A}$ by running Algorithm 2 for $T \in \mathbb{N}$ asynchronous steps with parameters $\mathcal{A}$, $K_i = 2^{i-1}$ and $B_i = B$. Then*

$$\mathbb{E}[f(\widehat{w}) - f(w^\star)] \leq \inf_{q \in (0,1]} \frac{DG\sqrt{32(1+2\tau_q)} + D\sigma\sqrt{48}}{\sqrt{qT}},$$

*where $\widehat{w} = \widehat{w}_I$ if $W$ is not empty, and $w_1$ otherwise.*

**Theorem 7.** *Let $f : \mathbb{R}^d \to \mathbb{R}$ be a $\beta$-smooth convex function admitting a minimizer $w^\star$, $g : \mathbb{R}^d \to \mathbb{R}^d$ be an unbiased gradient oracle of $f$ with $\sigma^2$-bounded variance, $T \in \mathbb{N}$ and $w_1 \in \mathbb{R}^d$. Let $\mathcal{A}(K)$ be SGD initialized at $w_1$ with stepsize $\gamma_k = \frac{1}{\beta}$. Let $W = (\widehat{w}_1, \ldots, \widehat{w}_I)$ be the outputs of $\mathcal{A}$ by running Algorithm 2 for $T \in \mathbb{N}$ asynchronous steps with parameters $\mathcal{A}$,*

$$K_i = 2^{i-1} \qquad and \qquad B_i = \max\left\{1, \left\lceil \frac{\sigma^2 K_i}{\beta^2 D^2} \right\rceil\right\}.$$

*Then*

$$\mathbb{E}[f(\widehat{w}) - f(w^\star)] \leq \inf_{q \in (0,1]} \frac{12(1+2\tau_q)\beta D^2}{qT} + \frac{\sigma D\sqrt{288}}{\sqrt{qT}},$$

*where $\widehat{w} = \widehat{w}_I$ if $W$ is not empty, and $w_1$ otherwise.*

### E.1. Proof of Theorem 6

If $W$ is empty, then $T < B$ (as $T_1 = 1$ and the condition $1 = t_1^{(1)} \leq t - d_t$ is always satisfied). Hence, as $T \geq 1$,

$$T < \frac{\sigma^2}{G^2},$$

and using the Lipschitz and diameter assumptions,

$$f(\widehat{w}) - f(w^\star) = f(w_1) - f(w^\star) \leq DG < \frac{D\sigma}{\sqrt{T}} \leq \frac{D\sigma}{\sqrt{qT}}$$

for all $q \in (0, 1]$. We proceed to the case where $W$ is not empty. Let $q \in (0, 1]$. By Lemma 2,

$$qT < 2(B + \tau_q)K_{I+1}.$$

If $B \leq \max\{1, \tau_q\}$,

$$qT \leq 2(1 + 2\tau_q)K_{I+1} \implies \frac{D^2(G^2 + \sigma^2/B)}{K_I} \leq \frac{8(1+2\tau_q)D^2G^2}{qT}.$$

If $B > \max\{1, \tau_q\}$, $\sigma > G$, $B = \lceil \sigma^2/G^2 \rceil$ and

$$qT \leq 4BK_{I+1} \implies \frac{D^2(G^2 + \sigma^2/B)}{K_I} \leq \frac{8BD^2(G^2 + \sigma^2/B)}{qT} \leq \frac{24D^2\sigma^2}{qT},$$

where the last inequality follows by $BG^2 \leq 2\sigma^2$. We are left with applying Lemma 3. To that end note that $\tilde{g}_k^{(I)}$ is the average of $B$ i.i.d. unbiased estimations of $\nabla f(\tilde{w}_k^{(I)})$, and by the linearity of the variance has variance bounded by $\sigma^2/B$. Thus, our selection of parameters enable us to use Lemma 6 and establish that

$$\mathbb{E}\left[f(\widehat{w}_I) - f(w^\star)\right] \leq \frac{2D\sqrt{G^2 + \sigma^2/B}}{\sqrt{K_I}} \leq \frac{DG\sqrt{32(1 + 2\tau_q)} + D\sigma\sqrt{96}}{\sqrt{qT}}.$$

As this holds for any $q \in (0, 1]$,

$$\mathbb{E}\left[f(\widehat{w}_I) - f(w^\star)\right] \leq \inf_{q \in (0,1]} \frac{DG\sqrt{32(1 + 2\tau_q)} + D\sigma\sqrt{96}}{\sqrt{qT}}. \qquad \square$$

### E.2. Proof of Theorem 7

If $W$ is empty, then $T < B_1$ (as $T_1 = 1$ and the condition $1 = t_1^{(1)} \leq t - d_t$ is always satisfied). Hence, as $T \geq 1$,

$$T < \frac{\sigma^2}{\beta^2 D^2},$$

and from smoothness,

$$f(\widehat{w}) - f(w^\star) = f(w_1) - f(w^\star) \leq \frac{\beta D^2}{2} < \frac{\sigma D}{2\sqrt{T}} \leq \frac{\sigma D}{2\sqrt{qT}}$$

for all $q \in (0, 1]$. We proceed to the case where $W$ is not empty. Let $q \in (0, 1]$. By Lemma 2,

$$qT < 2(B_{I+1} + \tau_q)K_{I+1}.$$

If $B_{I+1} \leq \max\{1, \tau_q\}$,

$$qT \leq 2(1 + 2\tau_q)K_{I+1} \implies \frac{1}{K_I} < \frac{4(1 + 2\tau_q)}{qT}.$$

If $B_{I+1} > \max\{1, \tau_q\}$,

$$qT \leq 4B_{I+1}K_{I+1} \leq \frac{8\sigma^2 K_{I+1}^2}{\beta^2 D^2} \implies \frac{1}{K_I} \leq \frac{\sigma\sqrt{32}}{\beta D\sqrt{qT}}.$$

We are left with applying Lemma 4. To that end note that $\tilde{g}_k^{(I)}$ is the average of $B_I$ i.i.d. unbiased estimations of $\nabla f(\tilde{w}_k^{(I)})$, and by the linearity of the variance has variance bounded by $\sigma^2/B_I$. Thus, our selection of parameters enable us to use Lemma 5 and establish that

$$\mathbb{E}\left[f(\widehat{w}_I) - f(w^\star)\right] \leq \frac{\beta D^2}{K_I} + \frac{2\sigma D}{\sqrt{B_I K_I}} \leq \frac{3\beta D^2}{K_I} \leq \frac{12(1 + 2\tau_q)\beta D^2}{qT} + \frac{\sigma D\sqrt{288}}{\sqrt{qT}}.$$

As this holds for any $q \in (0, 1]$,

$$\mathbb{E}\left[f(\widehat{w}_I) - f(w^\star)\right] \leq \inf_{q \in (0,1]} \frac{12(1 + 2\tau_q)\beta D^2}{qT} + \frac{\sigma D\sqrt{288}}{\sqrt{qT}}. \qquad \square$$

## F. Proofs of Section 4

### F.1. Proof of Lemma 2

Let $i \in [I + 1]$ and $k \in [K_i]$. Let $n(i, k)$ be the number of rounds with delay less or equal $\tau_q$ at times $t_k^{(i)} \leq t < t_{k+1}^{(i)}$ for $k < K_i$ and $t_{K_i}^{(i)} \leq t < t_1^{(i+1)}$ for $k = K_i$. Assume by contradiction that $n(i, k) > B_i + \tau_q$, and let $S = (a_1, \ldots, a_{n_{i,k}})$ be the

rounds of the $(i, k)$ iteration with $d_{a_i} \leq \tau_q$ ordered in an increasing order. For any $i > \tau_q$, as the sequence is non-decreasing and $a_1 \geq t_k^{(i)}$,

$$a_i - t_k^{(i)} \geq a_i - a_1 \geq \tau_q \geq d_{a_i},$$

which means that the inner "if" condition is satisfied and contradicts the condition of the while loop, $b \leq B_i$, since there is at least $B_i + 1$ rounds with $i > \tau_q$. Thus, $n(i, k) \leq B_i + \tau_q$ for all $i \in [I + 1]$ and $k \in [K]$. As each $B_i + \tau_q$ rounds with delay less or equal $\tau_q$ during the $i$'th iteration must account for a response to $\mathcal{A}$, and as we have at least $\lceil qT \rceil$ such rounds in total,

$$qT < \sum_{i=1}^{I+1} K_i(B_i + \tau_q),$$

otherwise $\widehat{w}_I$ will not be the last produced $\widehat{w}_i$. Hence, as $(B_i)_i$ is non-decreasing and $(K_i)_i$ is a geometric series,

$$qT < (B_{I+1} + \tau_q) \sum_{i=1}^{I+1} K_i \leq 2(B_{I+1} + \tau_q)K_{I+1}. \qquad \square$$

### F.2. Proof of Theorem 3

For completeness, we restate each part of Theorem 3 as a full theorem and provide the proofs.

**Theorem 8** (Theorem 3 (i)). *Let $f : \mathbb{R}^d \to \mathbb{R}$ be a $\beta$-smooth function lower bounded by $f^\star$, $g : \mathbb{R}^d \to \mathbb{R}^d$ an unbiased gradient oracle of $f$ with $\sigma^2$-bounded variance, $T \in \mathbb{N}$ and $w_1 \in \mathbb{R}^d$. Let $\mathcal{A}(K)$ be SGD initialized at $w_1$ with stepsize $\frac{1}{\beta}$. Let $W = (\widehat{w}_1, \ldots, \widehat{w}_I)$ be the outputs of $\mathcal{A}$ by running Algorithm 2 for $T \in \mathbb{N}$ asynchronous rounds with parameters $\mathcal{A}$,*

$$K_i = 2^{i-1} \qquad and \qquad B_i = \max\left\{1, \left\lceil \frac{\sigma^2 K_i}{2\beta F} \right\rceil\right\},$$

*for some $F \geq f(w_1) - f^\star$. Then*

$$\mathbb{E}\left[\|\nabla f(\widehat{w})\|^2\right] \leq \inf_{q \in (0,1]} \frac{24(1 + 2\tau_q)\beta F}{qT} + \frac{24\sigma\sqrt{\beta F}}{\sqrt{qT}},$$

*where $\widehat{w} = \widehat{w}_I$ is $W$ is not empty, and $w_1$ otherwise.*

The proof of Theorem 3 (i) is already provided in Section 4.

**Theorem 9** (Theorem 3 (ii)). *Let $f : \mathbb{R}^d \to \mathbb{R}$ be a $\beta$-smooth convex function admitting a minimizer $w^\star$, $g : \mathbb{R}^d \to \mathbb{R}^d$ be an unbiased gradient oracle of $f$ with $\sigma^2$-bounded variance, $T \in \mathbb{N}$ and $w_1 \in \mathbb{R}^d$. Let $\mathcal{A}(K)$ be accelerated SGD initialized at $w_1$ with parameters $\alpha_t = \frac{2}{t+1}$ and $\gamma_t = \gamma t$ where $\gamma = \frac{1}{4\beta}$ and $D \geq \|w_1 - w^\star\|$. Let $W = (\widehat{w}_1, \ldots, \widehat{w}_I)$ be the outputs of $\mathcal{A}$ by running Algorithm 2 for $T \in \mathbb{N}$ asynchronous rounds with parameters $\mathcal{A}$,*

$$K_i = 2^{i-1} \qquad and \qquad B_i = \max\left\{1, \left\lceil \frac{\sigma^2 K_i(K_i + 1)^2}{12\beta^2 D^2} \right\rceil\right\}.$$

*Then*

$$\mathbb{E}\left[f(\widehat{w}) - f(w^\star)\right] \leq \inf_{q \in (0,1]} \frac{192(1 + 2\tau_q)^2\beta D^2}{q^2 T^2} + \frac{48\sigma D}{\sqrt{qT}},$$

*where $\widehat{w} = \widehat{w}_I$ if $W$ is not empty, and $w_1$ otherwise.*

*Proof of Theorem 9.* If $W$ is empty, then $T < B_1$ (as $T_1 = 1$ and the condition $1 = t_1^{(1)} \leq t - d_t$ is always satisfied). Hence, as $T \geq 1$,

$$T < \frac{\sigma^2}{3\beta^2 D^2} < \frac{\sigma^2}{\beta^2 D^2},$$

and from smoothness,

$$f(\widehat{w}) - f(w^\star) = f(w_1) - f(w^\star) \le \frac{\beta D^2}{2} < \frac{\sigma D}{2\sqrt{T}} \le \frac{\sigma D}{2\sqrt{qT}}$$

for all $q \in (0, 1]$. We proceed to the case where $W$ is not empty. Let $q \in (0, 1]$. By Lemma 2,

$$qT < 2(B_{I+1} + \tau_q)K_{I+1}.$$

If $B_{I+1} \le \max\{1, \tau_q\}$,

$$qT \le 2(1 + 2\tau_q)K_{I+1} \implies \frac{1}{K_I(K_I + 1)} < \frac{1}{K_I^2} \le \frac{16(1 + 2\tau_q)^2}{q^2T^2}.$$

If $B_{I+1} > \max\{1, \tau_q\}$,

$$qT \le 4B_{I+1}K_{I+1} \le \frac{2\sigma^2 K_{I+1}^2(K_{I+1} + 1)^2}{3\beta^2 D^2} \implies \frac{1}{K_I(K_I + 1)} \le \frac{\sigma\sqrt{32/3}}{\beta D\sqrt{qT}} \le \frac{4\sigma}{\beta D\sqrt{qT}}.$$

We are left with applying Lemma 5. To that end note that $\tilde{g}_k^{(I)}$ is the average of $B_I$ i.i.d. unbiased estimations of $\nabla f(\tilde{w}_k^{(I)})$, and by the linearity of the variance has variance bounded by $\sigma^2/B_I$. Thus, our selection of parameters enable us to use Lemma 5 and establish that

$$\mathbb{E}\left[f(\widehat{w}_I) - f(w^\star)\right] \le \frac{4\beta D^2}{K_I(K_I + 1)} + \frac{4\sigma D}{\sqrt{3B_I K_I}} \le \frac{12\beta D^2}{K_I(K_I + 1)} \le \frac{192(1 + 2\tau_q)^2\beta D^2}{q^2T^2} + \frac{48\sigma D}{\sqrt{qT}}.$$

As this holds for any $q \in (0, 1]$,

$$\mathbb{E}\left[f(\widehat{w}_I) - f(w^\star)\right] \le \inf_{q \in (0,1]} \frac{192(1 + 2\tau_q)^2\beta D^2}{q^2T^2} + \frac{48\sigma D}{\sqrt{qT}}. \qquad \square$$

