# OpenReview forum: "Faster Stochastic Optimization with Arbitrary Delays via Adaptive Asynchronous Mini-Batching"
_ICML.cc/2025/Conference — ICML 2025 poster_

### Official Review · Reviewer_31cT · 2025-03-12

**Overall Recommendation:** 3

**Summary:**

The paper introduces a framework for asynchronous stochastic optimization that leverages quantile delays instead of traditional average delay measures. It presents a black-box conversion that transforms any standard stochastic first-order method into an asynchronous version with only simple analyses of classical methods. It also provides an adaptive procedure for asynchronous non-convex smooth and convex Lipschitz problems that automatically tunes to the best quantile without needing prior knowledge of the delay distribution, yielding accelerated rates.

## update after rebuttal

While this is a theoretical paper, numerical validation could have been included in this submission to further demonstrate the usability and robustness of their framework as a 'black-box' conversion to any first-order stochastic methods. Therefore, although the analysis of quantile delays sounds new to me, I will keep my current rating 3 due to the lack of experiments.

**Claims And Evidence:**

The submission provides rigorous theoretical proofs for convergence rates based on a chosen quantile of delays and accelerated convergence rates for convex smooth problems. In addition, the adaptive variant in Algorithm 2 does not require pre-specified delay bounds and automatically adapts to the best quantile in hindsight.

While the theoretical evidence is solid within its assumptions, the paper could further strengthen its theoretical findings with empirical validations or simulation studies for practical benefits.

**Essential References Not Discussed:**

Some recent related works in stochastic optimization with delayed updates are missing, e.g., [1,2].

[1]. Adibi, A., Dal Fabbro, N., Schenato, L., Kulkarni, S., Poor, H.V., Pappas, G.J., Hassani, H. and Mitra, A. Stochastic approximation with delayed updates: Finite-time rates under markovian sampling. In International Conference on Artificial Intelligence and Statistics (pp. 2746-2754), 2024.
[2]. Dal Fabbro N, Adibi A, Poor HV, Kulkarni SR, Mitra A, Pappas GJ. Dasa: Delay-adaptive multi-agent stochastic approximation. In IEEE Conference on Decision and Control (pp. 3889-3896). 2024.

**Experimental Designs Or Analyses:**

N/A

**Methods And Evaluation Criteria:**

The theoretical results in this paper are built heavily upon the theorem with classical stochastic methods from existing works (Lan, 2012, 2020), but it makes sense to modify the existing results to be adapted to asynchronous optimization under arbitrary delays.

**Other Comments Or Suggestions:**

It would be beneficial to discuss potential extensions of the framework to weak smooth conditions (such as [1]), which have wide applications in deep neural network training. In addition, addressing Markovian noise that leads to biased gradient oracles, often seen in reinforcement learning, would be valuable.

[1]. Zhang, J., He, T., Sra, S. and Jadbabaie, A., Why Gradient Clipping Accelerates Training: A Theoretical Justification for Adaptivity. In International Conference on Learning Representations, 2020.

**Other Strengths And Weaknesses:**

Strengths:
1. The paper offers a novel perspective by replacing average delay measures with quantile delays, which provides a more robust framework for asynchronous optimization.
2. It presents detailed and rigorous convergence proofs, extending classical results to handle arbitrary delays and even achieving accelerated rates in convex smooth settings.
3. The adaptive procedure that automatically tunes to the best quantile without prior knowledge is innovative and has potential for broad applicability in distributed systems.

Weaknesses:

1. This work is entirely theoretical; the lack of empirical evaluation limits the practical effectiveness and robustness of Algorithms 1 and 2 in real-world scenarios.

**Questions For Authors:**

1. Do the authors plan to include empirical validations, and can you share any preliminary results?
2. Does the method in this paper cover heavy-tailed or non-stationary delay distributions?

**Relation To Broader Scientific Literature:**

The paper makes the following key contributions to the asynchronous optimization literature:
- Replace average delay measures with quantile delays for more robust convergence guarantees.
- Propose a black-box conversion that adapts standard stochastic methods to asynchronous settings.
- Introduce an adaptive mechanism that tunes automatically to the best quantile delay without prior knowledge.
- Achieve accelerated convergence rates in convex smooth settings, addressing gaps left by prior works such as Cohen et al. (2021) and Mishchenko et al. (2022), and relating to recent adaptive methods, e.g., Tyurin (2024).

**Theoretical Claims:**

I checked the proofs of main theorems in the main body and didn't observe any technical errors.

---

> ### Author Rebuttal · Authors · 2025-03-31
>
> Thanks for the feedback—please see below our responses to the main points you raised.
>
> > “This work is entirely theoretical; the lack of empirical evaluation limits the practical effectiveness and robustness of Algorithms 1 and 2 in real-world scenarios.”
>
> Our work is primarily theoretical, focusing on providing improved convergence guarantees for asynchronous stochastic optimization and improving our understanding of this fundamental problem. That said, we agree with the reviewer that a numerical evaluation comparing our methods to classical asynchronous algorithms can further strengthen our results. Since the rebuttal period this year is extremely short, we will only be able to complete this for the final version. Specifically, our plan is to implement a simulated environment of delays, and to conduct a small-scale experiment with synthetic data, allowing for a larger number of simulated machines, as well as a larger experiment (with a deep NN on a standard benchmark) with a smaller number of simulated machines.
>
> > “Do the authors plan to include empirical validations, and can you share any preliminary results?”
>
> See comment above.
>
> > “Does the method in this paper cover heavy-tailed or non-stationary delay distributions?”
>
> Note that we actually consider arbitrary sequences of delays that need not come from a probability distribution. Therefore we do support any kind of distribution (as a special case), including heavy tailed and/or non-stationary distributions.
>
> Thanks for pointing out additional references and further suggestions.

---

### Official Review · Reviewer_P5Nu · 2025-03-13

**Overall Recommendation:** 3

**Summary:**

This paper studies the convergence of asynchronous stochastic methods, and the key differences with the literature include 1) a relatively general algorithm framework; 2) a new model/characterization of the delays, which, compared to delay models like maximum/average delay, can characterize the delays better. They derive convergence results in terms of their delay model, and show the advantage through comparison with the literature.

**Claims And Evidence:**

Yes, the claims are supported by their theory.

**Essential References Not Discussed:**

No.

**Experimental Designs Or Analyses:**

No experiments.

**Methods And Evaluation Criteria:**

Didn't propose any new method.

**Other Comments Or Suggestions:**

I was not aware of some issues pointed out by Reviewer A1zA. Now I have read them and agree with his concern.

However, the q-quantile delay model used in this paper is new to me and is a better delay model compared to the "upper bound of delay" used in most existing works. Convergence analysis under the new model can help to reflect the effect of delay distribution rather than the maximum delay. Therefore, this paper may have general meaning to the community of asynchronous optimization.

**Other Strengths And Weaknesses:**

1. Strength: They consider general asynchronous stochastic methods under a new delay model which is much better than average/maximum delay. They derived strong results, especially Theorem 3 which shows that the algorithm can adapt to $q$ in the $q$-quantile delay model to achieve faster convergence. This is impressive.

2. Weakness:

- I would expect a theoretical bound $\bar{\tau}_q$ on $\tau_q$. This is because that $\bar{\tau}_q$ is used to determine $B$ in Algorithm 1, while $B$ may affect $\tau_q$. Therefore, without a theoretical bound, $\bar{\tau}_q$ may be difficult to determine.

- The adaptivity result in Theorem 3 is impressive, but I still don't understand how this adaptivity is achieved. I would expect the authors to explain more.

- The literature review is slightly narrow. Specifically,  many works consider the setting where each agent has a private local cost function, while they are not surveyed. I understand that the focus of this paper does not include that setting, but since this paper considers "asynchronous" methods, I would encourage the authors to survey papers in that category. To list a few:

[A] Mishchenko K, Iutzeler F, Malick J, et al, A delay-tolerant proximal-gradient algorithm for distributed learning, International conference on machine learning. PMLR, pp. 3587-3595, 2018.

[B] Soori S, Mishchenko K, Mokhtari A, et al, DAve-QN: A distributed averaged quasi-Newton method with local superlinear convergence rate, International conference on artificial intelligence and statistics. PMLR, pp. 1965-1976, 2020.

[C] Wu X, Magnusson S, Feyzmahdavian H R, et al, Delay-adaptive step-sizes for asynchronous learning, International Conference on Machine Learning, pp. 24093-24113, 2022.

Also, there are several papers that allow for arbitrary delays or attempt to adapt to the real delay patterns, such as [C] and

[D] Hannah, Robert, Fei Feng, and Wotao Yin. "A2BCD: Asynchronous acceleration with optimal complexity." International Conference on Learning Representations. 2019.

**Questions For Authors:**

Does Theorem 1 allow for the same parameter range of the algorithm $A(\sigma, K)$? I ask this question because many methods such as delayed gradient descent requires the step-size to be inversely proportional to the maximum delay to guarantee convergence.

**Relation To Broader Scientific Literature:**

Limited.

**Theoretical Claims:**

No.

---

> ### Author Rebuttal · Authors · 2025-03-31
>
> Thanks for the review and strong support! Please see below our responses to the main points you raised.
>
> > “I would expect a theoretical bound $\bar{\tau}_q$ on $\tau_q$. This is because that $\bar{\tau}_q$ is used to determine $B$ in Algorithm 1, while $B$ may affect $\tau_q$. Therefore, without a theoretical bound, $\bar{\tau}_q$ may be difficult to determine.”
>
> $\tau_q$ may range from $0$ to $\Omega(T)$, depending on the adversary determining the delays, as we do not impose any assumptions on their distribution. Consequently, Algorithm 1 requires external knowledge ($\bar{\tau}_q$) about the observed delays. This is analogous to the use of the average delay (or an upper bound on it) in previous work, except that quantiles are less sensitive to outliers.
>
> > “The adaptivity result in Theorem 3 is impressive, but I still don't understand how this adaptivity is achieved. I would expect the authors to explain more.”
>
> The somewhat surprising adaptivity stems from two key components. First, (synchronous) SGD can be used with a large batch size—determined by problem parameters—with minimal degradation in performance, even though the number of update steps is reduced, as long as the noise term dominates the error. Second, the sweep of batch sizes bypasses the discrepancy between the number of asynchronous updates that are not filtered (which is unknown) and the number of updates to $\mathcal{A}$. This doubling trick, at the cost of a small constant, enables us to “simulate” knowledge of the number of updates $\mathcal{A}$ will receive.
>
> > “Does Theorem 1 allow for the same parameter range of the algorithm $\mathcal{A}(\sigma,K)$? I ask this question because many methods such as delayed gradient descent requires the step-size to be inversely proportional to the maximum delay to guarantee convergence.”
>
> This is a good distinction. There are no constraints on the parameter range of $\mathcal{A}$ that stem from the asynchronous nature of the problem. The only effect of asynchronous gradients on $\mathcal{A}$ is the number of update steps, which is unknown in advance and thus requires a bound $\bar{\tau}_q$ or a batch size sweep.
>
> Thank you for pointing out additional relevant works. We will expand the literature review to include these settings in the final version.

---

### Official Review · Reviewer_d4yS · 2025-03-14

**Overall Recommendation:** 3

**Summary:**

N/A

**Claims And Evidence:**

N/A

**Essential References Not Discussed:**

N/A

**Experimental Designs Or Analyses:**

N/A

**Methods And Evaluation Criteria:**

N/A

**Other Comments Or Suggestions:**

N/A

**Other Strengths And Weaknesses:**

I do not believe I am sufficiently qualified to review this paper. However, I can make two observations:

The algorithms are poorly written, to the point of being nearly incomprehensible. It seems quite difficult to claim an acceleration in convergence without demonstrating it through a simulation study.

Concerning the rate, since I am not competent, I'll follow the other reviewers' opinions.

**Questions For Authors:**

N/A

**Relation To Broader Scientific Literature:**

N/A

**Theoretical Claims:**

N/A

---

> ### Author Rebuttal · Authors · 2025-03-31
>
> Thank you for your feedback and for your transparency regarding your familiarity with the field in relation to the review.
>
> > “The algorithms are poorly written, to the point of being nearly incomprehensible.”
>
> The algorithms aggregate gradients for mini-batching while filtering stale ones based on the delay. Algorithm 2 employs an additional outer loop to determine the optimal batch size, which is unknown in advance, using the standard doubling technique.
>
> > “It seems quite difficult to claim an acceleration in convergence without demonstrating it through a simulation study.”
>
> Accelerated SGD and accelerated rates are standard terms associated with Nesterov’s seminal accelerated gradient method and its stochastic variants, which achieve convergence rates of $O(1/T^2 + \sigma / \sqrt{T})$, without necessarily a connection to empirical performance.

---

### Official Review · Reviewer_Tmk7 · 2025-03-17

**Overall Recommendation:** 3

**Summary:**

The authors consider the problem of stochastic optimization with delays. Concretely, they minimize an objective function $f:\mathcal{W} \to\mathbb{R}$ for a convex set $\mathcal{W} \subseteq \mathbb{R}^d$. They consider access to a stochastic unbiased gradient oracle with variance $\sigma^2$. Additionally, at each round $t$, the gradient oracle provides the stochastic gradient at $w_{t-d_t}$ where the delay sequence $d_t$ can be arbitrary. The goal is to obtain optimization algorithms that obtain best convergence rates after $T$ rounds in terms of the delay distribution.


The authors consider arbitrary delay distribution, with no assumptions on the delay, while existing works make several simplifying assumptions on delay distribution (fixed delay, bounded delay, fixed delay per machine or knowledge of delay distributions).


Under arbitrary delays, the authors provide a method (Algorithm 1), where given a black-box algorithm for stochastic optimization, given an upper bound $\bar{\tau}_q \geq \tau_q$, they select the corresponding batch size $B = \max (1,\bar{\tau}_q )$ and the number of iterations for the algorithm $K = qT / (1 + 2\bar{\tau}_q ) $.

Using this they run the black-box algorithm, to obtain the best possible convergence rates in terms of $\tau_q$ for smooth non-convex, smooth convex, and non-smooth convex objectives. See Table 1 for the exact rates. Here, $q\in (0,1)$ denotes the quantile and $\tau_q$ denotes the delay quantile values. Crucially, these are much better than existing works that can only do this for $\tau_{avg}$ or the average delay.


Further, to eliminate the dependence on knowledge of the upper bound of $\tau_{q}$, they use a doubling trick in Algorithm 2, to run several copies of Algorithm 1, with geometrically increasing number of rounds in each copy and increasing batch-sizes determined by the corresponding function class. Again, applying this to all function classes above, they obtain an $\inf_{q\in (0,1)}$ for their convergence rates achieving the best possible(Theorem 3 and Table 1).


Finally, they show that vanilla SGD with fixed step size without modifications or additional assumptions will always incur the maximum delay in Theorem 4.

**Claims And Evidence:**

- **Core-idea for black-box conversion** : The core-idea behind their Algorithm 1 is to construct a mini-batch of $B$ gradients in an asynchronous fashion. They run $K$ steps of the algorithm with batch size $B$. In each step, they wait until $B$ stochastic gradients have been obtained to update the model. By choosing $B$ carefully, they can control both the number of steps and the delay in steps in terms of $\tau_q$. They state this in Lemma 1 and provide the proof just after it.
- **Core-idea for anytime algorithm**: To extend Algorithm 1 to Algorithm 2, they use the doubling trick on number of rounds $K_i$ for the $i^{th}$ instantiation of Algorithm 1. This allows them to obtain a similar bound on the batch size $B_i$, number of steps of black-box algorithm $K_i$, number of rounds $T$ and delay quantile $\tau_q$ for any $q\in (0,1)$ in Lemma 2. The proof is simple but insightful. To go from here to the convergence rates for any function class, they plug in the batch size in any convergence rate, then find its optimal value to recover the best possible rate for the function class in terms.
- **Lower-bound and additional results**: The lower bound  as well as the suboptimality of average delay versus that of best case quantile are shown by constructing arbitrarily bad delay distributions.

**Essential References Not Discussed:**

I think the authors might want to motivate the doubling trick as it has been used previously in online learning for anytime algorithms, eg -(van Erven et al 2011). Apart from that the literature review is pretty thorough.

**References**
- (van Erven et al 2011) Adaptive Hedging. NeurIPS.

**Experimental Designs Or Analyses:**

See Methods and Evaluation Criteria.

**Methods And Evaluation Criteria:**

The authors provided **no experiments**. As the goal of this paper is to propose an algorithm, I request the authors to provide some experiments for their algorithm in practice. Ideally one synthetic example(quadratics with gaussian noise) and 1 real world neural network example on any bad delay distribution where $\tau_{avg}/\tau_{median}$ is large would do.

One issue with several stochastic optimization algorithms, even those in (Lan 2020), are that the optimal step sizes depend on problem parameters like smoothness, or suboptimality gap. So, when implementing algorithms in practice, especially on NNs, one has to resort to tuning. In the case of Algorithm 2 here, the batch size $B_i$ also depend on problem parameters, so the authors should show if a tuning procedure alongwith some growth conditions on $B_i$ is sufficient in practice. For instance, for non-convex smooth objectives, the authors can choose to find the value of $\frac{\sigma^2}{2\beta F}$ by tuning.

**Other Comments Or Suggestions:**

- **Presentation**: I think certain parts of the main paper can be moved to the appendix and an experiment section and conclusion can be included. For instance, in Line 376-382 right column, is a standard property of smoothness and just a reference for it, like Nesterov's book can be provided. Also, the proof for the lower bound can be moved to the appendix. Apart from this, footnote 2 on page 4 is incomplete, there are two "the" in the first line of Page 8 and, on the sample page, "Lemma 3" is applied, but it hasn't been defined what this Lemma states.

**Other Strengths And Weaknesses:**

None.

**Questions For Authors:**

- **Lower bound**: I see that (Tyurin 2024) has a lower bound for delay distributions in their Theorem 1. Does the lower bound hold here, as this does not seem to be a zero-respecting algorithm. Also, I understand that it is concurrent work, but I request the authors to provide a more detailed comparison.
- **Delay distributions independent of stochastic gradients**: In Line 188-189, the authors assume that delay distributions are independent of stochastic gradients. Are there cases in distributed optimization where this is violated, for instance where machines with a bad delay and gradient noise, but the average gradient noise over all machine is small? Also, do existing works like (Tyurin \& Richtarik 2023) handle these cases?

**Relation To Broader Scientific Literature:**

The key contribution is to provide a generic recipe for conversion of arbitrary optimization algorithms to handle arbitrary delays by mini-batching. The insight is quite novel, and turns a hard problem, that people can solve for specific delay distributions using complicated algorithms, into a simple one.

**Theoretical Claims:**

- All the theoretical claims are correct and in fact have simple proofs that the authors have explained very clearly.
- I would like to add that both Algorithm 1 and 2 are quite simple and can be applied to any black-box algorithm, which makes the method quite novel and insightful.

---

> ### Author Rebuttal · Authors · 2025-03-31
>
> Thanks for the feedback—please see below our responses to the main points you raised.
>
> > “The authors provided **no experiments**. As the goal of this paper is to propose an algorithm, I request the authors to provide some experiments for their algorithm in practice.”
>
> Our work is primarily theoretical, focusing on providing improved convergence guarantees for asynchronous stochastic optimization and improving our understanding of this fundamental problem. That said, we agree with the reviewer that a numerical evaluation comparing our methods to classical asynchronous algorithms can further strengthen our results. Since the rebuttal period this year is extremely short, we will only be able to complete this for the final version. Specifically, in agreement with your suggestions, our plan is to implement a simulated environment of delays, and to conduct a small-scale experiment with synthetic data, allowing for a larger number of simulated machines, as well as a larger experiment (with a deep NN on a standard benchmark) with a smaller number of simulated machines.
>
> > “One issue with several stochastic optimization algorithms, even those in (Lan 2020), are that the optimal step sizes depend on problem parameters like smoothness, or suboptimality gap. So, when implementing algorithms in practice, especially on NNs, one has to resort to tuning. In the case of Algorithm 2 here, the batch size $B_i$ also depend on problem parameters, so the authors should show if a tuning procedure alongwith some growth conditions on $B_i$ is sufficient in practice. ”
>
> Compared to Algorithm 1, Algorithm 2 performs an outer sweep over different batch sizes. This addition is made to identify the optimal batch size, as the number of non-stale updates is unknown without prior knowledge of the delays. In a tuning scenario, one can directly tune the batch size, resulting in a simpler asynchronous algorithm that aggregates gradients for a mini-batch synchronous algorithm (essentially Algorithm 1 with a tuned batch size). We will include your suggestion in the experiments we will complete for the final version.
>
> > “**Lower bound**: I see that (Tyurin 2024) has a lower bound for delay distributions in their Theorem 1. Does the lower bound hold here, as this does not seem to be a zero-respecting algorithm. Also, I understand that it is concurrent work, but I request the authors to provide a more detailed comparison.”
>
> The lower bound from the concurrent work (Tyurin 2024) applies, in their specific delay model,  to the algorithms we discuss. The main distinction between our paper and (Tyurin 2024) lies in the computational model: they introduce a framework that deviates significantly from the arbitrary delay models considered in prior work (e.g., Cohen et al., 2021; Mishchenko et al., 2022; Koloskova et al., 2022). While their model is general, it is also somewhat abstract, resulting in somewhat more complex guarantees compared to the (arguably) more intuitive quantile-based convergence guarantees we provide. On the other hand, (Tyurin 2024) also establishes the optimality of filtering stale gradient technique with a well tuned batch size by proving a lower bound for zero-respecting first-order algorithms. We will include a more detailed comparison in the revision.
>
> > “**Delay distributions independent of stochastic gradients**: In Line 188-189, the authors assume that delay distributions are independent of stochastic gradients. Are there cases in distributed optimization where this is violated, for instance where machines with a bad delay and gradient noise, but the average gradient noise over all machine is small? Also, do existing works like (Tyurin & Richtarik 2023) handle these cases?”
>
> The scenario you refer to falls within the setting of asynchronous optimization with heterogeneous data, where different workers have access to different samples. Some works address this setting but require additional assumptions, such as fixed delays per machine (Theorem A.4 of Tyurin & Richtarik, 2023) or certain distributional assumptions (Assumption 2 of Mishchenko et al., 2022). Supporting this case without additional assumptions is not feasible, as the data on slower machines must be accessed sufficiently many times for effective minimization. This could indeed be a valuable extension, but one that probably deserves a separate study.
>
> Thanks for pointing out typos and further suggestions.

---

> > ### Comment · Reviewer_Tmk7 · 2025-04-01
> >
> > Thanks for the detailed response. I do not have any other questions.

---

### Official Review · Reviewer_A1zA · 2025-03-25

**Overall Recommendation:** 1

**Summary:**

This paper proposes an asynchronous mini-batch black-box algorithm that aggregates asynchronously computed stochastic gradients as input to any stochastic-gradient-type optimization algorithm. In contrast to performing biased update using stale stochastic gradients, the proposed algorithm adaptively aggregates delayed gradients according to the delay length $d_t$. Analysis is provided to demonstrate the effect of batch size $B = \max \\{1, \bar{\tau}_q\\}$ and its dependence to the delay quantile $\tau_q$ on the convergence rate for multiple kinds of objective functions. Furthermore, another variant of algorithm that adopts increasing batch size is proposed, suggesting that the increasing batch size would achieve a tighter convergence bound that is optimal over the delay quantile $\tau_q$.

**Claims And Evidence:**

As discussed below under "Theoretical Claims".

**Essential References Not Discussed:**

No

**Experimental Designs Or Analyses:**

- This paper lacks numerical experiments.
- It is not shown how to choose the step size $\beta$ and batch size sequence $B_i$ in practice. For instance, in order to achieve the optimal upper bound suggested in Theorem 3, one must know the optimal $q^\star$ and correspondingly $\tau_{q^\star}$ in order to choose $B_i, K_i$ that satisfy the conditions required in Theorem 3.

**Methods And Evaluation Criteria:**

# Regarding Algorithm 1
- What does "Play $w_t = \tilde{w}_k$" means?
- Since $w_{t-d_t} = \tilde{w}_k$, at the iteration $k$-th of algorithm $\mathcal{A}$, $\mathcal{A}$ always receives a mini-batch of stochastic gradients evaluated on $\tilde{w}_k$. From here, for simplicity consider when $\mathcal{A}$ is SGD, it seems that Algorithm 1 is only a synchronous SGD that waits for $B$ mini-batches from multiple workers at every iteration. I found it confusing as it differs from the usual asynchronous gradient paradigm. It would be more clear if the authors could expand on the discussion of how the asynchrony in the proposed algorithm can achieve acceleration against synchronous SGD, e.g., in consideration of the larger batch size $B = \max \\{ 1, \bar{\tau}_q \\}$ used in Algorithm 1.

**Other Comments Or Suggestions:**

- The claim of this paper would be more convincing if the authors can present numerical experiments on extremely large $\tau_{\rm avg}$ and show the benefit of Algorithm 1 against classical asynchronous algorithm such as [[Arjevani et al., 2020]](https://proceedings.mlr.press/v117/arjevani20a/arjevani20a.pdf).
- According to Table 1 in the non-convex setting, for large enough $T$ such that the $\mathcal{O}(1/\sqrt{T})$ term dominates the error bound, the proposed convergence rate $\mathcal{O}(\sigma / \sqrt{qT})$ would be slower than the existing $\mathcal{O}(\sigma / \sqrt{T})$ rate for $q \in [0,1]$.

**Other Strengths And Weaknesses:**

As discussed below.

**Questions For Authors:**

No.

**Relation To Broader Scientific Literature:**

The proposed algorithm is orthogonal to classical asynchronous gradient algorithm such as [[Arjevani et al., 2020]](https://proceedings.mlr.press/v117/arjevani20a/arjevani20a.pdf), in the sense that the proposed algorithm avoids applying biased gradient steps by filtering and buffering the delayed gradients.

**Theoretical Claims:**

# Regarding the proof of Lemma 1
- It is not intuitive to see how $a_i - a_1 \ge \tau_q$ is obtained in line 310.
- Similarly, it is not clear how the argument "since there are at least $B+1$ rounds with $i > \tau_q$." is obtained in line 314.
- By the notation $\mathcal{A}(\sigma, K)$, I assume algorithm $\mathcal{A}$ queried and received $K$ stochastic gradients, as stated in Algorithm 1 because $k$ the iteration counter increases only after line 266. Therefore, it is not clear how do we introduce another $K'$ in line 300 and what is the meaning behind assuming $K' < K$.
- It is not explained how the second inequality in line 324 is obtained.

Explanation regarding the above ambiguities shall be included in the proof.

# Regarding Theorem 4
- The proof of Theorem 4 considered the mini-batching algorithm, which is not the vanilla asynchronous SGD algorithm [[Stich & Karimireddy, 2020]](https://jmlr.csail.mit.edu/papers/volume21/19-748/19-748.pdf) [[Arjevani et al., 2020]](https://proceedings.mlr.press/v117/arjevani20a/arjevani20a.pdf) as claimed in the first paragraph of Section 5.
- I suggest the authors to provide references / proof for the case of smaller $\eta$, i.e., when $\eta \leq 6 / \beta (1 + \tau_{\rm max})$, as claimed in the paragraph of line 390.

---

> ### Author Rebuttal · Authors · 2025-03-31
>
> Thanks for the feedback. Due to space constraints, we provide our responses to the main points below.
>
> > “What does “Play $w_t=\tilde{w}_k$” means?”
>
> In the asynchronous paradigm we consider, at each step $t$ the algorithm plays a point $w_t$ (essentially the current model at time $t$). This line means that the point the algorithm selects is $\tilde{w}_k$.
>
> > “Since $w_{t-d_t}=\tilde{w}_k$, at the iteration $k$-th of algorithm $A$…always receives a mini-batch…how the asynchrony in the proposed algorithm can achieve acceleration against synchronous SGD…”
>
> Consider a scenario with $M$ workers, where one is significantly slower while the remaining workers are fast (e.g., three times as fast). Synchronous mini-batch SGD with batch size $M$ would wait for the slow worker at each iteration. In contrast, our algorithm with batch size $M$ leverages the fast workers to compute the one missing gradient, achieving a 1.5x speedup.
>
> > “Explanation regarding the above ambiguities shall be included…”
>
> We will provide a more detailed version of the proof of Lemma 1 in the final version. Throughout the proofs, we treated the quantile delays ($\tau_q$) as integers, which can be done without loss of generality because (A) the delays are all integers, so $\lfloor\tau_q\rfloor$ is also a $q$-quantile, and (B) smaller delay is better. So, we can always use $\lfloor \tau_q \rfloor$ instead of $\tau_q$. We will state it more clearly in the final version---thanks for pointing this out. Below are more details on the specific transitions the review mentioned.
>
> > “...how $a_i-a_1\leq\tau_q$ is obtained in line 310.”
>
> As the integer sequence $a_1,..,a_{n(k)}$ is increasing, $a_i\geq a_1+(i-1)$. As $i>\tau_q$, $i\geq \tau_q+1$ (these are integers), so $a_i\geq a_1+\tau_q$.
>
> > “...how the argument "since there are at least $B+1$ rounds with $i>\tau_q$." is obtained in line 314.”
>
> We assumed by contradiction that $n(k)>B+\tau_q$, where $n(k)$ is the number of rounds with delay less or equal $\tau_q$. Working with integers, $n(k)\geq B+\tau_q+1$. So the last $B+1$ of these will have indices $i>\tau_q$.
>
> > “By the notation $\mathcal{A}(\sigma,K)$, I assume algorithm $A$ queried and received $K$ stochastic gradients…how do we introduce another $K’$...”
>
> We differentiate between $K$, which is the number of updates $A$ needs to produce an output, and $K’$, the actual number of updates received. $K’$ cannot be larger than $K$ due to the loop structure, but we need to prove that $T$ asynchronous steps are enough to produce $K$ updates to $A$, which is ensured by proving that $K’\geq K$.
>
> > “...how the second inequality in line 324 is obtained.”
>
> The second inequality is a part of Lemma 1. What we prove in the Lemma is that if this inequality holds, then $K’\geq K$.
>
> > “I suggest the authors to provide…proof for the case of smaller $\eta$...as claimed in the paragraph of line 390.”
>
> We will include the proof for this case and are happy to provide further elaboration if requested by the reviewer.
>
> > “The claim of this paper would be more convincing if the authors can present numerical experiments…”
>
> Our work is primarily theoretical, focusing on providing improved convergence guarantees for asynchronous stochastic optimization and improving our understanding of this fundamental problem. That said, we agree with the reviewer that a numerical evaluation comparing our methods to classical asynchronous algorithms can further strengthen our results. Since the rebuttal period this year is extremely short, we will only be able to complete this for the final version. Specifically, our plan is to implement a simulated environment of delays, and to conduct a small-scale experiment with synthetic data, allowing for a larger number of simulated machines, as well as a larger experiment (with a deep NN on a standard benchmark) with a smaller number of simulated machines.
>
> > “The proof of Theorem 4 considered the mini-batching algorithm, which is not the vanilla asynchronous SGD algorithm [Stich & Karimireddy, 2020] [Arjevani et al., 2020] as claimed in the first paragraph of Section 5.”
>
> The vanilla asynchronous SGD algorithm is SGD which performs update steps in an asynchronous manner. The asynchronous SGD algorithm presented in [Stich & Karimireddy, 2020] is the instance of vanilla asynchronous SGD with constant delay. To accommodate arbitrary delays, previous works (e.g., Mishchenko et al., 2022) extended asynchronous SGD to general delays. Theorem 4 follows the definition of asynchronous SGD with arbitrary delays.
>
> > “According to Table 1…for large enough $T$…$O(\sigma/\sqrt{qT})$ would be slower than the existing $O(\sigma/\sqrt{T})$ rate….”
>
> As we remark in lines 111-113 and in Section C, for $q=0.5$, our new rates are of the same order or better than those of previous work. The reason is that $\tau_q\leq 2\tau_{avg}$ when $q=0.5$. As the median is a more robust quantity to outliers than the average, our bounds are preferable.

---

> > ### Comment · Reviewer_A1zA · 2025-04-04
> >
> > Thank you for your response, especially your explanation on the proof details, which will improve the readability of this paper when included in the main text. Below are my comments on addressing your response.
> >
> > > Consider a scenario with $M$ workers, where one is significantly slower while the remaining workers are fast (e.g., three times as fast). ... fast workers to compute the one missing gradient, achieving a 1.5x speedup.
> >
> > Your explanation adds a lot value towards understanding Algorithm 1. I suggest this idea to be presented in the main paper. In this sense, the actual wall clock time spent to run $T$ asynchronous rounds in Algorithm 1 is the same as asynchronous SGD in prior works, and a potentially faster convergence is claimed by running Algorithm 1.
> >
> > > As we remark in lines 111-113 and in Section C, for $q = 0.5$, our new rates are ... our bounds are preferable.
> >
> > I respectfully disagree with the authors that the proposed new rates are of the same order or better than those of previous work. Here is my argument:
> > - I agree with the authors that when comparing the coefficients of the higher order term, i.e., the $\mathcal{O}(1/T)$ error term,  $\inf_q \frac{1 + \tau_q}{q T}$ is better than the prior SOTA term $\frac{1 + \tau_{\rm avg}}{T}$.
> > - However, when an optimization algorithm is ran for large enough $T$, the slower converging error term, e.g., variance error at the rate of $\mathcal{O}(1/\sqrt{T})$, dominates the convergence bound and the effect of $\mathcal{O}(1/T)$ error term vanishes.
> > - When comparing the lower order term, the proposed rate $\frac{\sigma}{\sqrt{qT}}$ is slower than the prior SOTA term $\frac{\sigma}{\sqrt{T}}$. For example, if $q = 0.5$, to find solution $\widehat{w}$ such that $\mathbb{E}[\\| \nabla f(\widehat{w}) \\|^2 ] \leq \epsilon$, there exists a small enough $\epsilon > 0$ such that the proposed algorithm is 2$\times$ slower than SOTA. If $q = 1$, the proposed lower order term matches with SOTA, but the proposed higher order term $\frac{1 + \tau_q}{qT} = \frac{1 + \tau_{\rm max}}{T}$ is larger than the SOTA higher order term $ \frac{1 + \tau_{\rm avg}}{T}$, therefore a weaker error bound.

---

> > > ### Author Response · Authors · 2025-04-05
> > >
> > > We appreciate the reviewer for the thoughtful engagement and additional valuable comments. We will of course incorporate these important discussions into the final version of the paper.
> > >
> > > Regarding the comparison of our bounds to SOTA: I believe we are actually in agreement with the reviewer. When we stated that our bounds are of the “same order or better than those of previous work”, we were referring to a comparison of rates up to numerical constants. It is indeed possible that our bound (for $q=0.5$) is up to a factor of 2 weaker than the SOTA bound in the regime the reviewer pointed out.
> > >
> > > We apologize for the lack of clarity in our earlier response and hope this properly addresses the reviewer’s remaining concern. We will ensure this point is clearly explained in the final version of the paper.

---

### Decision · Program_Chairs · 2025-05-01

**Decision:**

Accept (poster)

**Comment:**

This paper studies the theoretical convergence rates of mini-batching asynchronous SGD algorithm via a blackbox model. Using a computation model that is similar to that of Rennala SGD, or so called "Fixed Compute Model", the paper first studies the effect of different delay quantiles $q$ on the convergence rate using a similar algorithm to Rennala SGD, then an adaptive method based on an exponentially increasing "epoch" size is studied, which achieves the convergence rate with optimal quantile without knowing $q$ in advance.

The 5 reviewers have a split in opinions at the initial review. A few reviewers found the proposed analysis to be new and interesting. On the other hand, reviewers also raised issues on the writing  quality, and on the significance of the new results in practice, especially in the absence of any numerical experiments to substantiate the authors' claim. Although the authors promised to include numerical experiments in the next revision, it is too early to judge if the latter can be successful. As such, the opinions have remained split after the rebuttal.

The paper has thus generated a lot of discussions during the AC Reviewer Discussion phase, which has cleared some confusions during the review phase. In my opinion, while I appreciate the theoretical contributions of proposing a simple scheme to achieve the best rate robust to different quantiles of delay, I found the paper's positioning and presentation to be suboptimal. As it stands, the main contribution for the paper is more of a theoretical one. The current presentation would have misled (some of) its readers into finding its values from the more practical perspective, yet this is one of its main weaknesses. At the moment due to the lack of numerical experiments, it cannot be judged if the proposed algorithms are practical.

As such, I am recommending a "Weak Accept" for the paper, where it can be included in the proceedings if there is sufficient space in the program.